# Engineering and exploiting synthetic allostery of NanoLuc luciferase

Zhong Guo[1,2,3], Rinky D. Parakra[4], Ying Xiong[5], Wayne A. Johnston[1,2,3], Patricia Walden[1,2,3], Selvakumar Edwardraja [6], Shayli Varasteh Moradi[1,2,3], Jacobus P. J. Ungerer[7,8], Hui-wang Ai[5], Jonathan J. Phillips[4,9 ✉] & Kirill Alexandrov[1,2,3,10 ✉]

Allostery enables proteins to interconvert different biochemical signals and form complex metabolic and signaling networks. We hypothesize that circular permutation of proteins increases the probability of functional coupling of new N- and C- termini with the protein's active center through increased local structural disorder. To test this we construct a synthetically allosteric version of circular permutated NanoLuc luciferase that can be activated through ligand-induced intramolecular non-covalent cyclisation. This switch module is tolerant of the structure of binding domains and their ligands, and can be used to create biosensors of proteins and small molecules. The developed biosensors covers a range of emission wavelengths and displays sensitivity as low as 50pM and dynamic range as high as 16-fold and could quantify their cognate ligand in human fluids. We apply hydrogen exchange kinetic mass spectroscopy to analyze time resolved structural changes in the developed biosensors and observe ligand-mediated folding of newly created termini.

[1] ARC Centre of Excellence in Synthetic Biology, Sydney, Australia. [2] Centre for Agriculture and the Bioeconomy, Queensland University of Technology, Brisbane, QLD 4001, Australia. [3] School of Biology and Environmental Science, Queensland University of Technology, Brisbane, QLD 4001, Australia. [4] Living Systems Institute, Department of Biosciences, University of Exeter, Exeter EX4 4QD, UK. [5] Center for Membrane and Cell Physiology, Department of Molecular Physiology and Biological Physics, Department of Chemistry, University of Virginia, 1340 Jefferson Park Avenue, Charlottesville, VA 22908, USA. [6] Australian Institute for Bioengineering and Nanotechnology, The University of Queensland, Brisbane, QLD 4072, Australia. [7] Department of Chemical Pathology, Pathology Queensland, Brisbane, QLD 4001, Australia. [8] Faculty of Health and Behavioural Sciences, University of Queensland, Brisbane, QLD 4072, Australia. [9] Alan Turing Institute, British Library 96, Euston road, London NW1 2DB, UK. [10] Centre for Genomics and Personalised Health, Queensland University of Technology, Brisbane, QLD 4001, Australia. ✉email: jj.phillips@exeter.ac.uk; kirill.alexandrov@qut.edu.au

Allosteric regulation of proteins endows them with the ability to interconvert different types of biochemical signals. It plays a central role in the emergence of complex and resilient metabolic and signaling networks in biological systems. While the thermodynamic and mechanistic models of allostery have significantly advanced in the last decade, our ability to construct artificial allosteric proteins with the desired input and output parameters remains limited[1–4]. This, in turn, hinders the advancement of the key goal of Synthetic Biology—construction of orthogonal modular signaling and metabolic circuits. The difficulties associated with the engineering of protein allostery are rooted in the fact that it is often based on subtle and highly cooperative structural rearrangements that convert binding events or covalent modifications into changes of biochemical activity[5]. Often the receptor and the reporter functions are structurally and functionally intertwined in the same protein domain, making re-purposing of the receptor function difficult.

The classical model of protein allostery can be interpreted as a specialized case of protein folding, where an inactive enzyme exists in a folding state structurally and energetically close to the active state[1]. Indeed, regulation of access to the active site of the archetypal and first-described "allosteric enzyme" (glycogen phosphorylase), involves a putative unfolding-folding structural transition of the 280 s loop in response to remote ligand binding[6,7]. Binding of the ligand contributes energy required to overcome the thermodynamic barrier and shift the conformational ensemble from inactive to active conformation. In principle ligand binding can both increase and decrease the free energy, thus achieving OFF to ON activity transition. In the classical model, binding of the ligand decreases the free energy of the system and results in the active conformation of the proteins[1]. In an alternative scenario, inactive disordered protein is structured following the binding of the ligand[5]. This mechanism may be more common than previously appreciated given that the phenomenon of intrinsically unfolded and marginally stable proteins appears to be widespread in biology[8]. At a high level of abstraction the same principle operates in protein biosensors based on split domains, switch modules with ligand-controlled alternative folding frames and chimeric systems where an introduced receptor domain creates a conditional structural disorder[9–15]. Despite the progress in construction of split biosensors, this approach is complicated by the issues related to the stability of split fragments, complementation rate and the reversibility of reporter reconstitution[16]. The complexity of structural rearrangements required for alternative folding frames to switch efficiently in ligand-dependent fashion complicates the design of such biosensors, and also requires extensive optimization to achieve fast response kinetics[17]. The biosensors based on regulatory domain insertion do not follow a general design principle and instead rely on identification of the receptor domains with large conformational changes and extensive optimization of linker length and structure[18,19]. Furthermore, the number of receptors that are sufficiently biophysically robust and undergo large structural transition upon ligand binding is quite low, limiting applicability of this approach[20]. Design of such domains has been largely limited to the redesign of natural chemically induced dimerization systems and is far from trivial[21]. Finally, biosensors based on ligand-induced association of multiple components are inherently concentration driven and can operate only within a limited concentration range of biosensor components. While introduction of caged variants of two component biosensors architecture alleviates this problem, it additionally complicates the biosensor architecture[22]. Furthermore, use of non-cooperative ligand binding domains are inherently fraught with a "hook effect" in which both low and high concentration of the analyte result in the same signal[23].

In this study, we test an approach, where entropically driven local unfolded inactive reporter domain can be structured and activated by ligand-induced non-covalent intramolecular cyclisation. We demonstrate that the developed biosensors can be engineered to recognize both proteins and small molecules, and can be used to for detection in biological samples with sensitivities exceeding the requirements for practical utilization. Using kinetic mass spectroscopy and hydrogen exchange, we provide evidence for the proposed molecular mechanisms. We propose that this approach may be applied to other protein scaffolds, thereby representing a path to standardized approach to construction of artificial allosteric switches.

## Results

**Theoretical considerations**. Entropically driven conditional local structural disorder was successfully used to construct and tune aptamer-based biosensors[24]. Application of this approach to proteins is complicated by the highly cooperative nature of protein three-dimensional structure and its relationship with activity. It is well established that protein circularization increases protein stability by decreasing the entropy introduced by the termini's and increasing the enthalpy of protein packing[25]. It was also shown that while maintaining overall protein fold, circular permutation of proteins may lead to a decrease of stability presumably through an increase in entropy of the new N-termini and C-termini[26]. We conjectured that extending these termini may favor solvated extended state of the protein thereby reducing its activity. It appears logical that constraining the conformational flexibility of the termini would decrease the entropy of the system by creating non-covalently cyclized structure with reduced structural disorder and increased activity. If the cyclisation event is mediated by a ligand, it would create an artificial allosteric protein. We set out to test this idea using a model luminescent protein.

**Construction of permutated NanoLuc luciferase**. Bioluminescent protein systems are used for numerous biological applications, ranging from highly sensitive biochemical assays to bioluminescence-based animal imaging. Recently NanoLuc luciferase (NanoLuc) from deep-water shrimp *Oplophagus gracilirostris* gained popularity due to its small size, enhanced stability, and >150-fold increase in luminescence over traditionally used enzymes[27]. A split version of NanoLuc has been used extensively to construct two component biosensors where ligand mediated scaffolding of a C-terminally truncated enzyme and a peptide(s) corresponding to the terminal β-strand(s) leads to reconstitution of enzymatic activity[28–30]. Recently, we demonstrated that insertion of calmodulin in the loop connecting β-strand 10 to the rest of the protein converts NanoLuc into a biosensor of calmodulin binding peptide[15]. The latter switch was used to construct a two component biosensor architecture that, at least in principle, could be used for detection of any analyte[15].

The ability of the last two β-strands of NanoLuc to reversibly associate with the rest of luciferase and efficiently restore its activity make the system a good test case for construction of a synthetic switch unit where ligand binding reduces local disorder and increases the enzyme's activity. To access terminal β-strands we fused the native N-terminus and C-terminus of NanoLuc with a flexible linker composed of glycine serine repeats and reintroduced them in the loop connecting the last β-strand with the rest of the protein at position 161, resulting in formation of a circular permutated NanoLuc (cpNanoLuc) (Fig. 1b and Supplementary Table 1). A similar circular permutation of NanoLuc was previously used to construct a protease-controlled OFF switch[29].

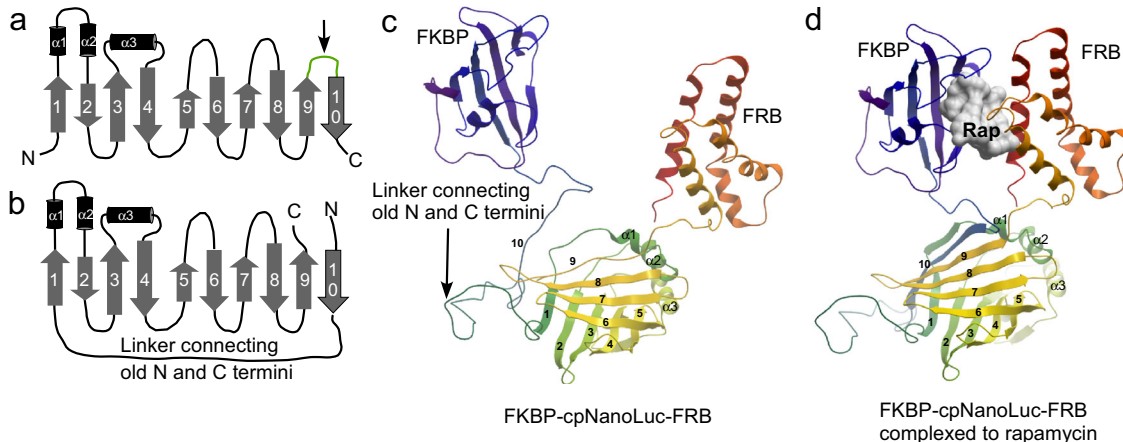

**Fig. 1 Engineering of circular permutated NanoLuc and thereon based biosensors. a** NanoLuc topology model[29] consisting of a 10-stranded β-barrel. The dissection point used to generate split NanoLuc is marked by an arrow. **b** Circular permutated version of NanoLuc shown as in **a**. The numbering of the β-strands is based on the wild type enzyme. **c** A model of rapamycin biosensor based on circular permutated NanoLuc colored spectrally from N-termini to C-termini (PDB: 5ibo). The elements of the secondary structure are labeled as in B and the β-strands 9 and 10 are displayed as lacking secondary structure. **d** As in **c** but in the presence of rapamycin (Rap) ligand (shown as white molecular surface). In the model the β-strands 9 and 10 occupying positions and structured as in the native enzyme. Source data are provided as a Source Data file.

We next sought to install bulky ligand binding domains at the new N-termini and C-termini in an effort to increase β-strand dissociation due to the action of solvation forces (Fig. 1c). This could be counteracted by ligand-mediated association of the binding domains limiting the conformational space that these β-strands can occupy. To test this idea, we fused the FRB and FKBP to the newly formed N and C terminus of the cp-NanoLuc and recombinantly produced the resulting fusion protein in *E. coli*. As can be seen in Fig. 2a, addition of rapamycin to the solution of the fusion protein resulted in the dose dependent and saturable six-fold increase in its luminescence. Noteworthy, the fit of the data indicated that the biosensor:rapamycin interaction displayed a $K_d$ value of 0.4 nM and the limit of detection of 50 pM. To confirm that the observed ligand-controlled activity switching was a consequence of circular permutation we fused FRB and FKBP to the N-termini and C-termini of wild type NanoLuc via linkers of comparable length and tested the recombinantly produced protein for rapamycin-dependent activity (Supplementary Table 1). Presence of rapamycin did not have appreciable influence on the activity of fusion protein thus providing support to the notion that circular permutation plays a key role in emergence of reversible local disorder (Supplementary Fig. 1D).

To perform the initial assessment of the possible changes in oligomerization state and conformation of the developed biosensors, we subjected rapamycin biosensors to size exclusion chromatography in the presence or absence of rapamycin. The biosensor eluted from the size exclusion column as a single peak of the expected molecular weight both in the presence and the absence of rapamycin (Supplementary Fig. 1E). This indicates that the protein is largely folded and monomeric in the absence of the ligand and that ligand's addition does not radically change its structure or oligomerization state. This is in line with the notion of local structural rearrangement of the termini controlled by the association of the ligand binding domains.

Encouraged by these results, we decided to test if the same approach can be used to construct biosensors of other small molecules. We chose the microcyclic immunosuppressant compound tacrolimus (FK-506) that is structurally related to rapamycin, but has much wider clinical use[31]. The main impediment to construction of a tacrolimus biosensor is the size of tacrolimus binding ternary complex, which is more than two times larger than the rapamycin binding FRB:FKBP complex. We

have solved this problem by using the structure of quaternary complex of Calcineurin A and Calcineurin B in complex with FKBP and FK506 (PDB:1TCO) to design a linker between Calcineurin A and Calcineurin B converting it to a single polypeptide[32]. This polypeptide was used to replace the FRB domain in the rapamycin biosensor yielding a putative tacrolimus biosensor (Supplementary Table 1). Titration of this biosensor with tacrolimus demonstrated dose dependent response, with the overall affinity for the drug of 0.4 nM and the limit of detection close to 50 pM (Fig. 2b). The Hill co-efficient of the titration data indicated the possible presence of positive cooperativity in the system (Supplementary Fig. 1A–C). We repeated the experiments using samples spiked with 50% serum or saliva. In both cases while we observed the reduction in overall signal, the response of the biosensor has not changed and the concentration of the drug in the sample could be reliably determined (Supplementary Fig. 2B, C). We also demonstrated that biosensors of rapamycin and tacrolimus were specific for their cognate drugs and no cross-activation was observed (Supplementary Fig. 2D and Supplementary Note 1). Fully activated biosensors retained approximately 20% of wild type NanoLuc's luminescence and could be dried down and re-hydrated without optimization (Supplementary Fig. 3B, C).

Testing general applicability of the developed biosensor architecture: next, we wanted to establish how the nature of the ligand binding domains affects biosensor's performance. Specifically, we wanted to find out if the developed biosensors can be used for detecting analytes other than small molecules. To this end we fused two VHH domains (also known as Nanobodies)[33] recognizing two non-overlapping epitopes of α-amylase to both N-termini and C-terminus of the developed biosensors. As can be seen in Fig. 2c, addition of α-amylase to the solution of VHH-cpNanoLuc-VHH fusion resulted in its dose dependent activation. The biosensor displayed excellent sensitivity with a limit of detection of 50 pM and over 4-fold dynamic range. To further validate the developed biosensor platform as generally applicable, we constructed a biosensor of human serum albumin (HSA) by attaching the previously described VHH domain at the N-terminus and a fragment of protein G at the C-terminus of the biosensor[15]. Like the α-amylase biosensor this HSA biosensor displayed 12-fold dynamic range and a limit of detection of 100 pM (Supplementary Fig. 4). These findings strongly support

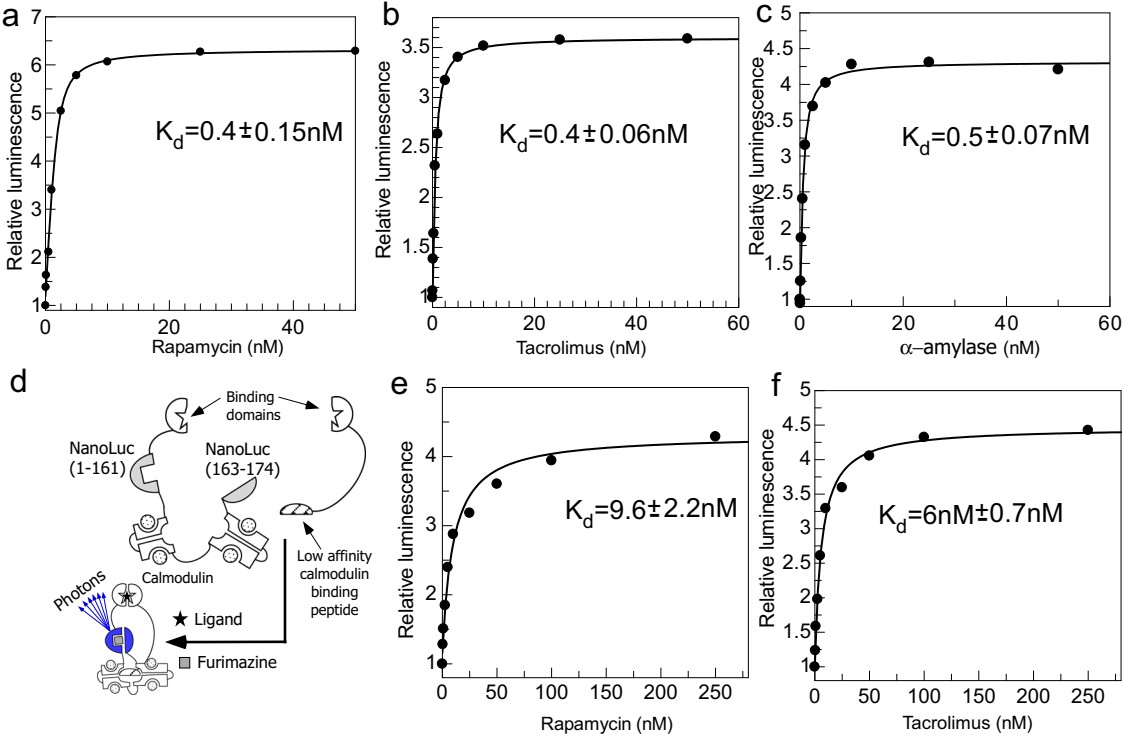

**Fig. 2 Changes in luminescence of NanoLuc-based biosensors upon titration with their cognate ligands. a** Titration of 200 µl solution of 1 nM Rapamycin biosensor supplemented with 0.25 µl furimazine stock solution in buffer containing 20 mM Tris-HCl pH 7.2, 100 mM NaCl with increasing concentrations of rapamycin. The data was fitted to a $K_d$ value of 0.4 nM. **b** As in **a** but using a chimeric protein where fusion of Calcineurin A and Calcineurin B replaces FRB domain and using tacrolimus (FK506) as a titrant. The data was fitted to a $K_d$ value of 0.4 nM. **c** As in **a** but using cpNanoLuc fused to anti α-amylase VHH domains for ligand recognition and α-amylase as a titrant. The data was fitted to a $K_d$ value of 0.5 nM. **d** A schematic representation of a two component Nanoluc based biosensor with calmodulin inserted into the loop connecting the last β-strand rendering the molecule inactive. Ligand-induced scaffolding of this chimeric unit with the calmodulin-binding peptide induces the conformation change of calmodulin and activation of the enzyme. **e** Activity of a two-component rapamycin biosensor shown in **d**. Here, a solution of 10 nM CaM-NanoLuc-FKBP and 30 nM FRB-CaM-BP in reaction buffer described in **a** was titrated with increasing concentrations of rapamycin. The data was fitted to a $K_d$ value of 9.6 nM. **f** As in **e** but using 10 nM CaM-NanoLuc-FKBP and 30 nM Calcineurin A and Calcineurin B fusion with calmodulin binding peptide and tacrolimus as titrant. The data was fitted to a $K_d$ value of 6 nM. Source data are provided as a Source Data file.

the notion that the developed architecture is generic and allows rapid construction of biosensors to small molecules and proteins given availability of selective binding domains.

In principle the developed biosensors should exist in an equilibrium with their cognate ligands, and hence decrease in the concentration of the latter should result in reversion of the biosensor's activation. Due to very high affinities of the developed biosensors and the corresponding slow off rates, this notion is almost impossible to test using solution assays. Therefore, we chose cpNanoLuc HSA biosensor for ligand dissociation experiments, as it has lowest affinity for its ligand among the biosensors generated. We immobilized the purified biosensor in the well of the microtiter plate and confirmed that the biosensor remained active and increased its luminescence following exposure to HSA (Supplementary Fig. 4B). We then extensively washed the wells with the buffer and retested the well's luciferase activity. As can be seen in Supplementary Fig. 4B, this significantly reduced the luciferase activity that we interpreted as dissociation of a large fraction of HSA ligand from the biosensor. Subsequent addition of HSA resulted in restoration of the luminescence. This strongly suggests that dissociation of the ligand from the biosensors leads to its de-activation via a shift to a more disordered state.

The developed biosensors demonstrated remarkably high affinities for their ligands and low detection limits. To understand the nature of this phenomenon, we compared the performance of single component cpNanoLuc biosensors with the two

component architecture developed previously[15]. Here, the activity of NanoLuc is modulated by calmodulin domain inserted in the loop connecting the last β-strand to the β-sheet. The biosensor is activated via ligand-mediated scaffolding with calmodulin binding peptide that induces conformational change in the chimeric reporter leading to its activation (Fig. 2d). We constructed two component NanoLuc biosensors of rapamycin and tacrolimus using the same binding domains as were previously used to construct circularly permutated single component variants and tested their ability to detect both compounds. Figure 2e, f shows that both biosensors responded to the cognate ligands but displayed 10–15 times lower affinity. This suggests that the high affinity of the cpNanoLuc biosensors for their targets is due to cooperativity arising from the binding domain being integrated in the same polypeptide chain.

Construction of the red shifted version of the cpNanoLuc biosensor: the developed biosensors are potentially suitable for in vivo application as they are fully genetically encoded and can be introduced into cells or animals by transfection. However, the blue emission of the enzyme with the maximum of 460 nm is away from the optimal optical window between 650 and 800 nm, and this limits its performance in blood and tissues[34]. We initially constructed a variant of cpNanoLuc, where eGFP was inserted into cpNanoLuc between N-terminal β-strand and the rest of the molecule. The resulting biosensors retained their functionality but displayed an emission maximum at 510 nm (Supplementary

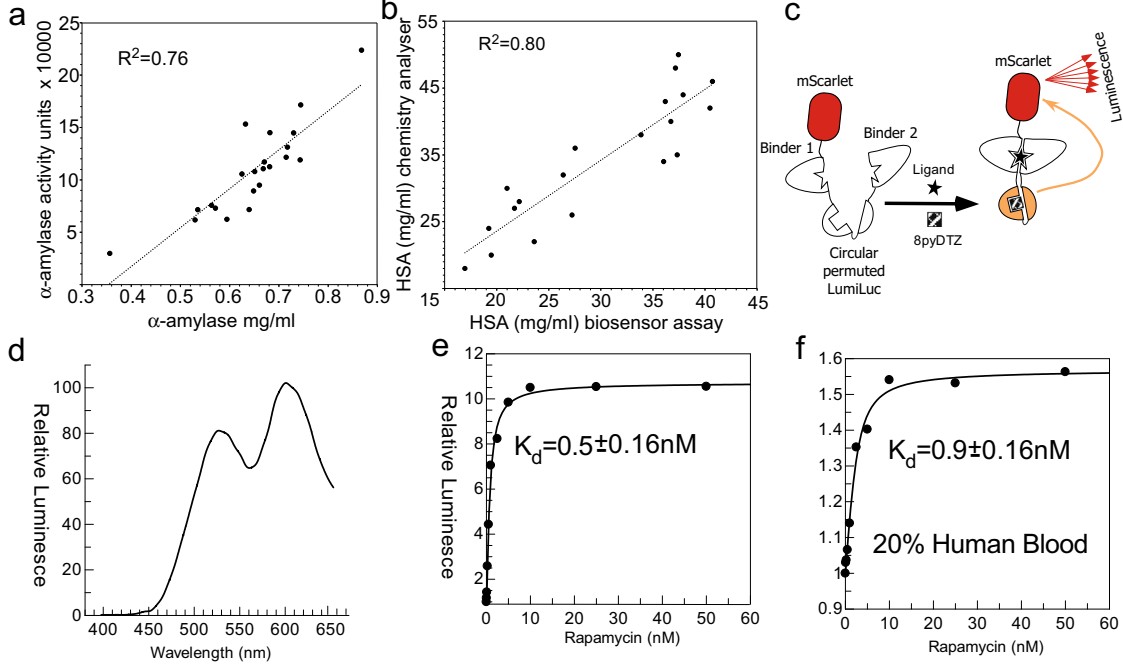

**Fig. 3 Quantification of biomarkers in human biological fluids using biosensors based on circularly permutated version of LumiLuc. a** Quantification of α-amylase in samples of human saliva using solution assay based on cpLumiLuc biosensor (*X*-axis) and its comparison with and its comparison with the clinical α-amylase activity assay (*Y*-axis). **b** Quantification of HSA concentration in serum of human donors using cpLumiLuc HSA biosensor (*X*-axis) or clinical chemistry analyser (*Y*-axis). **c** Schematic representation of the BRET biosensor based on cpLumiLuc. **d** Emission scan of the solution of 1 nM of LumiLuc biosensor, 10 nM Rapamycin, 30 μM 8pyDTZ. **e** titration of 1 nM solution of LumiLuc biosensor and 30 μM 8pyDTZ with increasing concentrations of rapamycin. The emission was collected at 595–620 nm. **f** As in **c** but using the sample supplemented with 20% lysed human blood. Source data are provided as a Source Data file.

Fig. 5). This, however, resulted only in a moderate improvement in biosensor's compatibility with blood samples.

Several studies reported variants of NanoLuc or/and its substrate furimazine that shift emission into the red part of the spectrum[35,36]. We set out to test whether these modifications are compatible with developed biosensor architecture and could be used to create cpNanoluc biosensors emitting in the red part of the spectrum. To this end, we introduced the twelve previously identified point mutations into cpNanoLuc rapamycin biosensor that resulted in a LumiLuc variant capable of accepting furimazine derivatives such as 8pyDTZ as its substrate[36]. We showed that the emission maximum of the biosensor shifted to 530 nm without significantly affecting the dynamic range (Supplementary Fig. 6A, B). We decided to test if such biosensors with red-shifted emission (thus expected to be less absorbed by biological fluids), could be used to quantify biomarkers in human samples. To this end, we constructed the LumiLuc version of the above-described biosensors of rapamycin, α-amylase and human serum albumin. As expected, the biosensors displayed shifted emission at 550 nm and their luminescence was dose-dependently increased upon ligand addition. Fitting the data showed that the modified biosensor had similar affinity to the ligand (Supplementary Fig. 6A, B). To develop an assay based on cpLumiLuc suitable for quantification of clinical biomarkers, we tested different buffer compositions to stabilize the cpLumiLuc signal and achieve glow-type bioluminescence[37] (Supplementary Fig. 6C, D). We then applied the developed assay to measure concentrations of α-amylase in human saliva and human serum albumin in human serum, and then compare the results to results obtained either with clinically used α-amylase activity assay or HSA quantification assay. As can be seen in Fig. 3a, b, in both cases we obtained very good correlation between methods. While for clinical deployment the accuracy of the assays must be further

improved, analysis of the data suggests that changes in luminescent intensity over time contribute to the standard error of the experiments (Supplementary Fig. 6E, F). This is a typical behavior of luciferase-based reporter systems and an elegant solution based on a reference luciferase with shifted wavelength was proposed recently providing a clear path to further improvement of the assay accuracy[38].

Next, we tested if further shifting of the emission wavelength could be achieved using bioluminescent resonance energy transfer (BRET) to a red fluorescent protein (Fig. 3a). We fused fluorescent protein mScarlet to either the C-terminus to the rapamycin biosensor or inserted it into the middle of the linker connected with the N-terminal and C-terminal part of cLumiLuc as described above for eGFP insertion. The recombinant variants of the resulting fusion assemblies demonstrated two emission peaks at 530 and 610 nm that correspond to the emission wavelength of LumiLuc and mScarlet (Fig. 3b). Interestingly, the dynamic range of the resulting biosensor increased to nearly ten-fold possibly indicating that structural rearrangements induced by ligand binding affected BRET efficiency in addition to the overall activity of the LumiLuc (Fig. 3c).

We next tested the ability of the biosensor to function in blood containing medium (i.e., that absorbs in the yellow and red part of the spectrum). To this end we repeated titrations of the developed rapamycin biosensor in the presence of blood, and demonstrated that addition of 20% of lysed blood did not significantly influence the obtained $K_d$ value (Fig. 3d).

Functional mechanism of cpNanoLuc biosensors. The results described above suggest that the developed biosensor architecture is generic and can be used to detect a broad range of analytes. However, its functional mechanism is inferred from static structural models and is speculative. Furthermore, it does not provide an explanation for the observed activation kinetics of the

biosensor or guidance for its further improvement (Supplementary Fig. 3). It also does not indicate which specific structural locations should be engineered to enhance sensitivity to even lower ligand concentrations. Hence, next we set out to gain insights into the functional mechanism of the developed biosensor architecture. Given the complexity and the size of the engineered fusion proteins, one method that is able to provide sufficient temporal but also structural resolution is hydrogen/deuterium-exchange mass spectrometry (HDX-MS)[39,40]. While HDX-MS can enable measurement of solvent accessibility and hydrogen-bonding at up to single amino acid resolution through soft-fragmentation techniques[41], here we chose to calculate a residue average of overlapping peptides at each position to produce estimates of deuterium labelling that were resolved at the per-amino acid level, using the heatmap function in DynamX (Waters). This is similar to the method of Keppel and Weis although the weighting of measurements is linear with peptide size[42]. This was used to develop a highly structurally-resolved model of the allosteric activation of the biosensor protein. As a test example, we chose the rapamycin biosensor that utilizes the smallest binding domains with a mass of 45 kDa. We decided to compare the hydrogen exchange rates at sites within this biosensor in the absence and in the presence of seven concentrations of rapamycin to correlate this with local structural changes (Fig. 4A). The rapamycin biosensor was at 8 μM for this experiment and so the rapamycin was added between 0–20 μM, i.e., up to a 2.5:1 stoichiometry. This ensured binding saturation (99% rapamycin-bound after 20× dilution with $D_2O$ buffer), given the $K_d$ estimate of 0.5 nM from a luminescence assay. Here, the hydrogen-exchange was measured as the extent of observed deuterium labeling with deuterated Tris buffered saline (TBS) at three mixing times, across three orders of magnitude. Notably, structural responses to ligand were observed at the fastest HDX mixing time (30 s—top panel of Fig. 4A) in the FKBP domain, with little significant change in any other part of the protein. At an intermediate mixing time (300 s) there were large amounts of protection seen throughout the FKBP domain as well as smaller differences in β-10 of the cpNanoLuc domain and in the FRB domain, especially in the α2–α3 loop. After longer exposure to deuterium labeling (3000 s), large changes in HDX are observed in every part of the protein, indicating that ligand binding affects structural dynamics throughout and so each part of the protein potentially impacts rapamycin sensitivity. When bound with saturating amounts of rapamycin (20 μM), both the FKBP and FRB domains show a similar very low fractional uptake of deuterium label at each time point measured (red traces in Fig. 4A). Yet in the unbound (apo) form of the protein they have strikingly different uptake of deuterium label (black traces in Fig. 4A). The unbound FKBP domain exchanges on average twice as much deuterium label as the FRB domain (average fractional uptake for all peptides at all time points 0.29 compared to 0.14). However, the average predicted intrinsic HDX rates are similar (1.8 compared to 2.0 s$^{-1}$), solvent accessible surface area (SASA) for the domains is similar (6086 compared to 6320 Å$^2$) and the surface burial by rapamycin is similar (532 compared to 736 Å$^2$)[43]. This indicates that there is a far greater stabilization of the FKBP domain upon rapamycin, likely with a large entropic component. Stabilization of the NanoLuc domain is observed upon rapamycin binding, but this effect is evidenced only at the longest HDX mixing time (3000 s), indicating the H-bond network of the two beta-sheets is intact, but involves dynamic instability in the apo form.

To characterize the distinct structural transitions that occur at equilibrium from apo-protein to the mature active enzyme-rapamycin complex, we made a quantitative analysis of the HDX data in response to ligand and at each amino acid in the protein

**Table 1 Example amino acids assigned to each cluster that shows a defined response to ligand, with parameters from fit to a dose-response curve.**

|  | Val2 | Arg57 | Ile91 | Phe99 | Ala127 | Val175 | Trp280 |
|---|---|---|---|---|---|---|---|
| $EC_{50}$ | n.d.[a] | 6.0 | 5.1 | 3.3 | 8.5 | 10.2 | 8.2 |
| $n_H$ | n.d.[a] | −2.5 | −2.3 | −1.2 | −1.6 | −2.5 | −2.5 |
| Cluster | 6 | 1 | 5 | 2 | 3 | 7 | 4 |

[a]Rapamycin-dependent response not detected.

for which we had data (Supplementary Figs. 7–8 and Supplementary Data 1). We sought to cluster coherent conformational changes that are stabilized by ligand binding. To achieve this, we calculated the sum of observed deuterium labeling for the three mixing times at each rapamycin concentration. These were then fitted to a dose-response model (examples of unprocessed peptide level data shown in Fig. 4B and averaged per amino acid data shown in S8). This yielded two coefficients to the model fit (equation S2)—the $EC_{50}$ midpoint and the Hill number, $n_H$, which represent the sensitivity and cooperativity respectively of each amino acid to rapamycin ligand. By clustering the amino acids according to these attributes, we identified those parts of the protein that have correlated behavior in response to ligand (Fig. 4B). Examples from each of the seven clusters are shown in Table 1 (note cluster 6 indicates no detectable rapamycin-dependent response). It is possible that the observed range of $EC_{50}$ values for amino acids that are perturbed upon rapamycin binding is an artefact of experimental factors, such as the limited D-labeling times. If real, then it would be indicative of a broad and rough energy landscape for conformers of the active ligand-bound state of the protein, as has been widely observed in natural systems, such as multistate functional selectivity in G-protein coupled receptors (GPCRs)[44–48] and the folding of linear repeat proteins[49–52]. There are confounding factors that would complicate and potentially alter this interpretation, such as conformers with higher or lower protection factors than the (un)bound states that are populated before or after on the pathway, or distortion of the apparent titration curve resulting from the HDX measurement error. Ultimately a kinetic model is preferred, though here that analysis would be under-determined. Nonetheless, this analysis provides considerable insight into the allosteric conformational changes of this biosensor.

The rapamycin biosensor protein has six clusters of amino acids that share a common response to ligand binding, with examples shown in Fig. 4C and Table 1; a seventh cluster (cluster 6) represents null data and those amino acids non-responsive to ligand, both of which were assigned an arbitrary $EC_{50}$ and Hill number of zero to enable inclusion in clustering. This results in clusters 6 and 2 (the amino acids with the most sensitive response to ligand and so the lowest $EC_{50}$ values) showing close linkage, but this is artefactual. Therefore, excluding cluster 6, there is common behavior in clusters 2, 5, and 1 and, to a much-reduced degree, with cluster 4 which includes much of the cpNanoLuc domain. However, clusters 7 and 3 are divergent, sharing little or no commonality. These clusters each correspond to amino acids in a specific location within the cpNanoLuc domain (3—glycine rich circular permutation linker; 7—α1 helix) which are the least sensitive to ligand, based on their high $EC_{50}$ values.

This collectively allows us to infer an equilibrium pathway of ligand-dependent activation (Fig. 5). The most sensitive ligand response (clusters 2 and 5) is from the two well-characterized rapamycin binding domains (FKBP and FRB), logically indicating that they bind first. Three possibilities rationally exist for this: (i) they bind ligand together (central pathway); (ii) FKBP domain

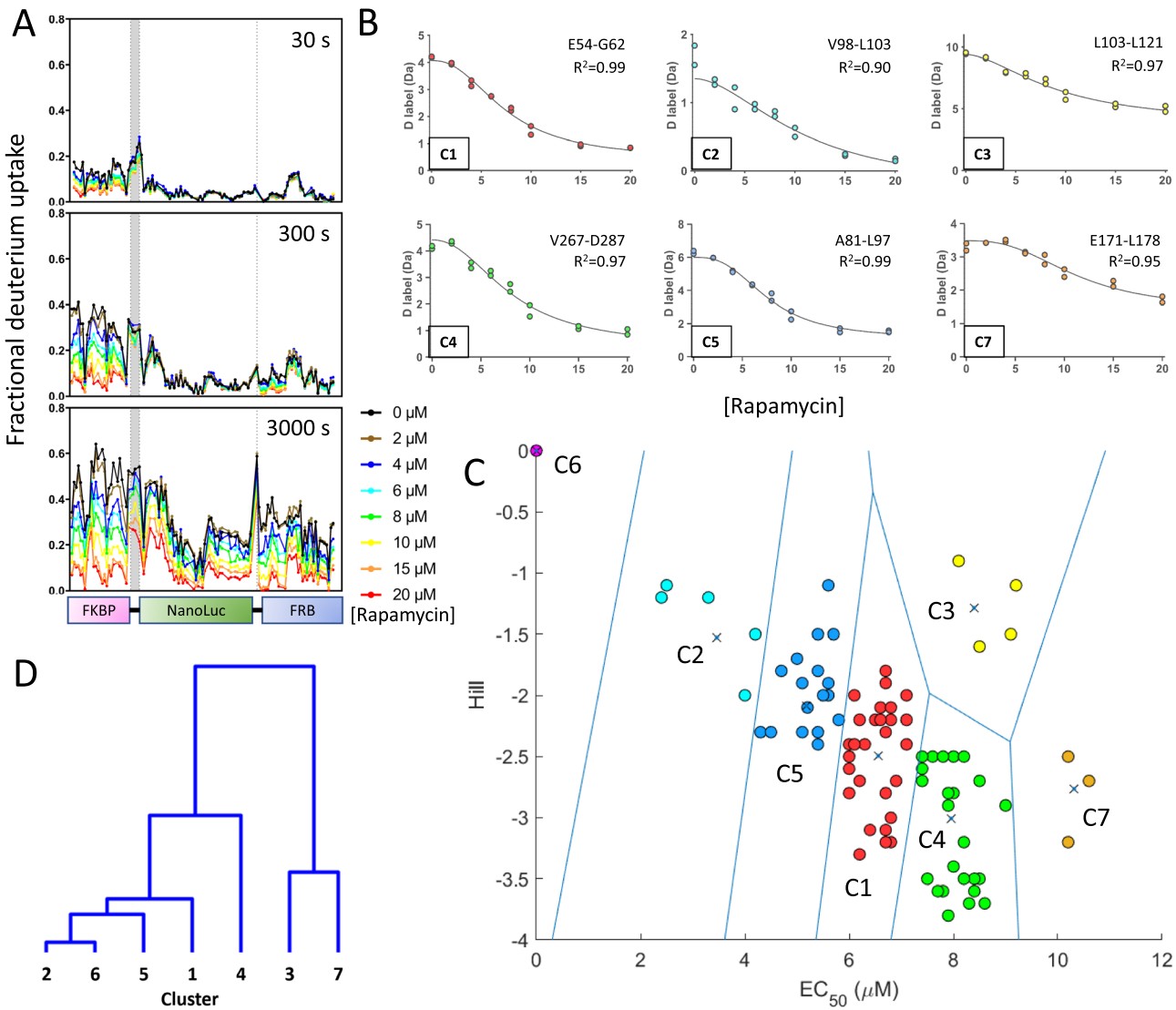

**Fig. 4 Binding of rapamycin induces significant orthosteric and allosteric changes in the structural ensemble of FKBP-cpNanoLuc-FRB with coherent behavior. A** Fractional deuterium labelling at three mixing times (30, 300, and 3000 s) as a function of rapamycin concentration (0, 2, 4, 6, 8, 10, 15, and 20 μM) binding to 8 μM FKBP-cpNanoLuc-FRB. Protein domains shown below on the *x*-axis. Gly-rich linkers between domains as dashed vertical lines. **B** Unprocessed HDX-MS labelling data at 3000 s mixing time as a function of rapamycin concentration are shown for representative peptides of each cluster (except C6 rapamycin-independent regions). Fit to a Hill equation (black trace). **C** Rapamycin dose-response of each amino acid in the biosensor protein (as defined by $EC_{50}$ and Hill coefficients from fit to Hill equation) was clustered with a k-means method. HDX-MS measurements were averaged per amino acid from peptide-level data and summed across all time points. Centroid of each cluster (C1–C7) denoted by *x*; Voronoi boundaries—blue lines. **D** Dendrogram of families from clustering in **C**. *Y*-axis—linkage (arbitrary units in Euclidean space). Source data are provided as a Source Data file.

binds first or (iii) FRB domain binds first. As the cluster contains amino acids spread across both FKBP and FRB domains, it is clear that either simultaneous FKBP/FRB binding occurs via the central pathway or that (ii) and (iii) occur as parallel pathways. Given the likely conformational degrees of freedom between the two binding domains, especially considering that afforded by the extraction of β-10, we propose that it is more probable the parallel pathways are populated than both domains bind simultaneously.

By clustering the data in this manner, certain features of the biosensor allosteric mechanism become clear (Fig. 5). Firstly, the data confirm a central part of the rapamycin biosensor design hypothesis, that the open (apo) conformation of the rapamycin biosensor does indeed have β-10 of the cpNanoLuc domain detached from the β-sheet. This is evidenced by two separate results: (i) the weak protection against hydrogen-exchange in β-10 (Val116-Cys122) when no rapamycin is present, which would

be unexpected for a strand within a stably hydrogen-bonded β-sheet and is not observed for the other strands here and (ii) β-10 is a member of cluster 1 (red), so is one of the most sensitive parts of the cpNanoLuc domain to respond to ligand on the allosteric equilibrium pathway—the majority of that domain has a higher $EC_{50}$ and is clustered separately (cluster 4—green). Together this supports an interpretation that it readily intercalates into the β-sheet on-pathway to rapamycin-induced activation, separately and in advance of the more extensive rearrangements in the rest of the cpNanoLuc domain. The final two clusters of perturbation on the equilibrium pathway of activation correspond to structural rearrangements in the glycine-rich circular permutation linker (cluster 3—yellow) and to the α-1 helix (cluster 7—orange). These are the least sensitive structural features to ligand and indicate precisely the amino acids that should be targeted in future engineering to optimize the sensitivity of the rapamycin biosensor

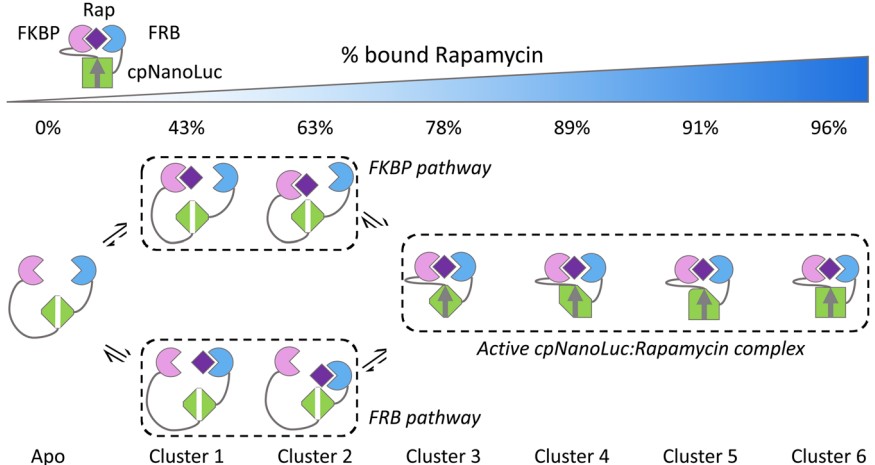

**Fig. 5 Model of the FKBP-cpNanoLuc-FRB conformational activation.** Rapamycin (purple) binds to the FKBP (pink) and FRB (blue) domains and stabilizes a "closed" form of the central cpNanoLuc (green) domain in which the β-10 strand has intercalated into the β-sheet. The centroid of each cluster has its midpoint at the given %bound rapamycin. Parallel pathways exist for ligand binding, though the central pathway (simultaneous FKBP/FRB binding to rapamycin) is improbable. Few structural changes were observed beyond 1:1 stoichiometry of binding where the biosensor is estimated 89% bound to rapamycin. β-10 strand (arrow) forms from FKBP-cpNanoLuc linker following rapamycin binding.

—in particular the α-1 helix region whose structural transition is significantly less sensitive than all other parts of the protein: $EC_{50}$ midpoint of the transition occurs 1.9 μM after the previous transition, which equates to an additional 25% ligand saturation. Importantly, these structural perturbations are sensitive to levels of rapamycin significantly greater than the apparent saturation of the biosensor; with a $K_d$ of 0.5 nM then full complexation (96%) would be achieved with 1:1.3 mixing at 0.38 μM, following 20× dilution into $D_2O$ buffer, indicating that they may be off-pathway effects, for example non-specific interaction with ligand that would require separate detailed investigation.

The results in Figs. 4 and 5 indicate that addition of a ligand induces structural changes in the FKBP and FRB domains that propagate to the β-strands of cpNanoLuc, resulting in reduction of their HDX labelling and ultimately that of almost the entire protein. We interpret this to mean that both N-terminal and C-terminal domains and the β-strands undergo reordering upon ligand binding, and that this may explain the large dynamic range of the constructed biosensors as two coordinated events (with the potential for frustration in the event of rapamycin binding to both FKBP and FRB simultaneously), required for reconstitution of activity. This arrangement can be interpreted as a logic AND gate with two, at least partially, independent switch inputs. The seeming existence of the parallel binding pathways also constitutes a logic OR gate, determined stochastically by the initial binding to either N-terminal or C-terminal domain. A previous study demonstrated the construction of NanoLuc tripart reconstitution system, where two last β-strands and the rest of NanoLuc molecule were scaffolded together by a ligand resulting in a switch with a large dynamic range[30].

## Discussion

Circular permutation of proteins has been used extensively to study mechanisms of protein folding and to engineer proteins with new functions[53,54]. Particularly in the case of fluorescent proteins newly introduced termini provided regulatory handles on protein activity, and led to the construction of a range of biosensors with various utilities[55,56]. The success of this approach can be rationalized from the evolutionary perspective, where the fitness selection has favored proteins that are functionally insulated from the active site to minimize effects of termini's

dissociation, degradation, or post-translational modifications. Relocation of the termini to a new position increases the likelihood of them to be associated with the rest of the molecule less tightly than in the case of native termini that were evolutionarily optimized to prevent protein unfolding. The solvation forces can induce termini-initiated local unfolding that interferes with the function of the proteins, thereby creating conditions for emergence of OFF states. In the current work we achieved control of NanoLuc activity by relocating its termini to the loop connecting the last β-strand to the rest of the protein. The HDX-MS experiments sugest that new N-termini and C-termini connecting the enzyme to the ligand binding domains are disordered in the ground state, leading to low activity NanoLuc that is enhanced up to 16-fold when ligand-mediated scaffolding occurs. This event can be viewed as a non-covalent protein cyclisation that reduces the disorder of the newly designed termini and increases the catalytic activity of the system. This provides a work around for the necessity to directly couple the ligand binding event to a defined conformational change, which complicates the design of integrated artificial allosteric switches. We expect that by modifying sequences of β-strands 9 and 10, as well as the length and composition of the linkers connecting the cpNanoLuc and binding domains it will be possible to modulate the dynamic range and response times of the system. An additional variable that is expected to influence the behavior of the developed switch is the length of the linker connecting original N and C termini.

Clearly, validation and assessment of the general nature of this phenomenon as well as the analysis of the governing forces (entropy, solvation, and steric effects) cannot be performed using a single permutation and would require a quantitative analysis of a library of circular permutated reporter domains, a larger set of ligand binding domains as well as additional biophysical methods. There is, however, additional evidence that the suggest that the observed phenomenon may not be unique to NanoLuc and other reporter domains could be converted into useful biosensors using a similar approach[57].

The intramolecular cyclisation and re-ordering mode could provide an explanation to the observation, why the system is so remarkably tolerant to the topology of the ligand binding domains and could be controlled by both small molecules and proteins. The presented single component biosensor architecture

can be targeted to structurally diverse analytes ranging from small molecules to biological heteropolymer thereby extending the family of NanoLuc-based biosensors[15,29,58–62]. This is particularly encouraging in application to small molecules where significant progress has been made in constructing chemically induced dimerization systems using a bottom-up approach[63]. The alternative approach is based on protein domains that undergo conformation change upon ligand binding, and developing binding domains that selectively recognize only the ligand-bound conformation[64–67]. Despite this progress binder development even for simpler targets such as proteins remains a challenge that requires considerable skills in the use of display systems in order to obtain binding domains to non-overlapping epitopes.

The presence of the ligand binding domains in *cis*- is also expected to create an avidity effect, where the high local concentration of both binders reduces the overall $k_{off}$ thereby increasing the affinity of the system.

The luminescent signal generated by the biosensor was further used to create its BRET variants based on mutant versions of the NanoLuc (LumiLuc)[36]. In this case, the emission of the biosensor was shifted to the red part of the spectrum using fused mScarlet fluorescent protein, allowing it to work effectively in blood containing samples. Furthermore, the BRET biosensor enables ratiometric, hue-based readout decreasing reliance on the absolute signal intensity. This enables the use of such biosensors for construction of simple microfluidic paper-based analytical devices that could be used for detection of specific antibodies in blood samples[68,69]. Further, semi-synthetic BRET biosensors showed promise in constructing miniaturized detection systems for small molecule drugs[70–72] and nucleic acids[73].

In conclusion, we propose that entropically driven local structural disorder may represents an alternative mechanism for construction of artificial allosteric systems with increased tolerance for the size and topology of the regulatory ligand. While much more comprehensive analysis is required to establish general applicability of this phenomenon, our results provide guidance on the design of such studies.

## Methods

**Ethical statement**. Authors confirm that this study complied with all relevant ethical regulations. Samples of human biological fluids were collected under the human ethics approval provided by Queensland University of Technology Human Research Ethics Committee (HERC) under approval number 1900000068. The approval is based on prior approval of the lead HERC of Royal Brisbane and Women's Hospital HREC/18/QRBW/226. The de-identified samples from routine diagnostic test runs were obtained from Pathology Queensland. The gender and age information were not recorded for this study. No compensation of the donors took place.

**Materials**. Rapamycin, tacrolimus, human α-amylase, human serum albumin, and BSA were purchased from Sigma-Aldrich. NanoLuc substrate Furimazine was purchased from Promega. The luciferase assay plate OptiPlate-96 HS were purchased from PerkinElmer.

**Expression vector construction and protein expression and purification**. The fusion open reading frames were generated from PCR products or gBlock synthetic fragments (Integrated DNA Technologies) by Gibson Assembly method according to the manufacturer's instruction (New England Biolabs) (see Supplementary Table 1 for the ORF sequences).

The biosensor components were cloned into pET28a and expressed in cytosol of *E.coli* BL21(DE3)RIL cells, except the proteins with VHH domains were expressed into periplasm with a signal peptide sequence. Proteins were purified by Ni-NTA chromatography, and the pooled protein-containing fractions were dialyzed against buffer containing 20 mM Tris/HCl pH 7.2, and 100 mM NaCl for 10 h. The constructs containing VHH domains were additionally purified by size exclusion chromatography on Superdex 200 column (GE Healthcare) to remove oligomers. The constructs had typical purity between 75 and 95%. The proteins were concentrated to approximately 100 μM concentration, snap frozen in liquid nitrogen and stored at −80 °C.

**Analysis of luminescent biosensors and processing of the titration data**. All luminescence measurements have been performed at TECAN plater reader (SPARK). In the assay the solution of protein biosensor was preincubated with the analyte at room temperature for 15 min and the luminescence reaction was initiated by the addition of the luciferase substrate. For furimazine, the luminescence was recorded at 445–470 nm; for 8pyDTZ the luminescence was recorded at 520–545 nm.

In order to obtain the $K_d$ of the interaction the absolute or relative luminescence values for the individual samples of titration experiments were plotted against the concentration of the ligand and the data were fitted to the explicit solution of the quadratic equation describing the E + S <> ES binding equilibrium, where $K_d$ is defined as

$K_d = [E] * [S]/[EL]$. $[E_0]$ and $[L_0]$ refer to the total enzyme and ligand concentration (free and bound) in the cuvette. Under these conditions the luminescence is described by Eq. 1

$$L = L_{obs(min)} + (L_{obs(max)x} - L_{obs(min)})/ * (([E_0] + [L_0] + K_d)/2 - ([E_0] + [L_0] + K_d)^2/4 - [E_0] * [L_0])^{1/2}/[L_0] \tag{1}$$

$L_{obs}$ represents the luminescence, while $L_{obs(min)}$ and $L_{obs(max)}$ refer to the minimal and maximal luminescence observed, respectively. A least-squares fit of the data to Eq. 1 using the software package Grafit 7.0 (Erithacus software) was used to determine the $K_d$ value.

**Calculation of Hill coefficient**. In order to calculate the Hill coefficient, we estimated the fraction of the luciferase biosensor that was bound by the analyte as the ratio of the luminescence signal changing (the difference of luminescence signal in the presence or absence of the analyte) at defined concentration of the analyte to the maximum luminescence at the saturated concentration of analyte. The concentration of free analyte was calculated based on the difference of the total concentration of analyte and the concentration of analyte, which bound to biosensor. The Hill coefficient was obtained by the linear fitting of the Hill plot.

**Size exclusion chromatography**. Size exclusion chromatography was performed on Superdex 75 10/300 GL column (GE Healthcare) equilibrated with buffer 20 mM Tris pH 7.2, 100 mM NaCl and in case of the complex with 5 μM rapamycin. For analysis 250 μl solution of cpNanoLuc Rapamycin biosensor at the concentration of 1 mg/ml with or without addition and 15 min pre-incubation of 1:2 molar ratio of rapamycin was injected onto the column. The column was developed with two column volumes of the same buffer at the flow rate of 0.5 ml/min and the absorbance at 280 nm was recorded. The column was calibrated using gel filtration protein standards (Sigma).

**Surface immobilization of and activity analysis of cpNanoLuc HSA biosensor**. The assay was performed in 96-well plates (OptiPlate-96 HS, Perkinelmer). The wells were incubated for 15 min with 200 μl solution containing 20 mM Tris/HCl pH 7.2, 100 mM NaCl, 1 mg/ml BSA and 1 nM cpNanoLuc biosensor. Subsequently HSA was added to the chosen wells to a final concentration of 100 nM. The plate was washed with 20 mM Tris/HCl pH7.2, 100 mM NaCl and after adding of 0.25 μl the NanoLuc substrate to the well the luminescence was recorded at 445–470 nm. The plate was washed six times with 200 μl assay buffer containing 20 mM Tris/HCl pH 7.2, 100 mM NaCl. After washing, 200 μl assay buffer was added to the plate's wells with or without 100 nM HSA. Following 1 h incubation the luminescence was measured again following the addition of 0.25 μl NanoLuc substrate. The luminescence signal of plastic absorbed biosensor represented about 1% of the input material.

**Deuterium exchange experiments and data analysis**. Hydrogen exchange was performed using an HDX Manager (Waters) equipped with an HDX2 CTC PAL sample handling robot (LEAP Technologies). Prior to HDX-MS experiments the purified FKBP-cpNanoLuc-FRB protein was diluted to 8 μM for mass spectrometry experiments using the 20 mM Tris-HCl, 100 mM NaCl pH 7.2. Except for the apo sample, this buffer also contained 0–20 μM rapamycin, to promote saturating conditions ($k_d = 1$ nM). Samples of FKBP-cpNanoLuc-FRB in protonated aqueous buffer were diluted 20-fold into deuterated buffer (20 mM Tris-HCl, 100 mM NaCl pD 7.2) at 20 °C, initiating hydrogen exchange. The protein was incubated for either 30, 300, or 3000 s in the deuterated buffer and at least two replicates were collected per condition. Hydrogen exchange was arrested by mixing 1:1 with pre-chilled quench buffer (8 M Urea, 100 mM potassium phosphate, pH 2.45 at 0 °C). The protein was then digested into peptides on a pepsin column (Enzymate, Waters) and the peptides were separated on a C18 column (1 × 100 mm ACQUITY UPLC BEH 1.7 μm, Waters) with a linear gradient of acetonitrile (3–40%) supplemented with 0.2% formic acid. Peptides were analysed with a Synapt G2-Si mass spectrometer (Waters). The mass spectrometer was calibrated with NaI/CsI calibrant in positive ion mode. A clean blank injection was run between samples to minimise carry-over. Peptide mapping, where peptides were identified by MS[E] fragmentation using eight collision energy ramps and with TWIMS ion mobility separation in N₂ gas, was performed prior to the hydrogen exchange experiments and analysed using ProteinLynx Global Server-PLGS (Waters) to positively identify peptides. Peptide mapping yielded coverage of 97% of N-terminally His-tagged

protein with a high degree of redundancy (3.53) (Supplementary Fig. 6). The data pertaining to deuterium uptake (data presented in Supplementary Fig. 7 and in supplementary HydrogenExchangeData file) were analysed and visualised in DynamX 3.0 (Waters), Matlab (Mathworks) and Prism (GraphPad Software, US). No correction was made for back-exchange.

Dose-response curves for fitting the sum of observed HDX deuterium label versus [rapamycin] per amino acid were done as follows: HDX data were resolved at the amino acid level by linearly weighting peptide level measurements (at all observed charge states; $z$) for exchangeable amide groups in DynamX 3.0 (Waters). These data were collected for seven [rapamycin] with a minimum of two replicates and normalized locally per amino acid. Mean and 1 s.d. were determined for each data point and these were fit automatically to the normalized Hill equation for inhibition (Eq. 2), using lsqcurvefit() in Matlab (Mathworks). The $EC_{50}$ and Hill number from these fits represent the sensitivity and cooperativity, respectively, of each amino acid to ligand. These were k-means clustered into groups and visualized on protein structural models in Pymol (Schrödinger).

$$Y = \frac{1}{\left(\frac{1 + x^{n_H}}{EC_{50}^{n_H}}\right)} \qquad (2)$$

**Analysis of cpNanoLuc tacrolimus biosensor specificity, activation rate and stability**. To determine the specificity of the biosensors 1 nM solution of tacrolimus biosensor either in buffer or in the presence of 200 molar excess of rapamycin (200 nM) or 1000 molar excess of cyclosporine A (1 μM) was titrated with increasing concentrations of tacrolimus. The data was fitted to a quadratic equation to obtain the apparent $K_d$ of the systems.

**Quantification of HSA in serum samples by LumiLuc based sensor**. To determine the concentration of human serum albumin in patient serum, the samples were diluted with the buffer containing 20 mM Tris pH 7.2 and 100 mM NaCl to 1:500 folds. The assays were performed in 200 μl formulated assay buffer A containing 1 mM CDTA, 0.5% Tergitol NP-40, 0.05% Antifoam 204, 150 mM KCl, 100 mM MES, pH 6.0, 1 mM DTT, and 35 mM thiourea in the presence of 1 μl diluted serum sample, 30 μM substrate 8pyDTZ, and 1 nM of LumiLuc based protein sensor[36]. The mixture was first incubated at 25 °C for 30 min without substrate. Upon addition of substrate the change of luminescence signal 520–545 nm was monitored with TECAN plater reader (SPARK). The signals then were used to determine albumin concentration by correlating it with the calibration curve. The calibration curve was obtained by titration of the sensors with the known amount of human serum albumin (Sigma-Aldrich).

**Quantification of HSA in human serum using Beckman Coulter Synchron DxC800 chemistry analyser**. The samples of human serum were analysed using bromcresol dye using the ALBm kit and according to the instructions of the manufacture.

**Quantification of α-amylase in saliva**. The saliva samples were diluted with buffer containing 20 mM Tris pH 7.2 and 100 mM NaCl to 1:25 folds. The assays were performed in 200 μl formulated assay buffer containing 1 mM CDTA, 0.5% Tergitol NP-40, 0.05% Antifoam 204, 150 mM KCl, 100 mM MES, pH 6.0, 1 mM DTT, and 35 mM thiourea in the presence of 1 μl diluted serum sample, 30 μM substrate 8pyDTZ and 1 nM of LumiLuc based protein sensor. The mixture was first incubated at 25 °C for 30 min without substrate. Upon addition of substrate the change of luminescence signal 520–545 nm was monitored with TECAN plater reader SPARK. The signals then were used to determine α-amylase concentration by correlating it with the calibration curve. The calibration curve was obtained by titration of the sensors with the known amount of human α-amylase (Sigma-Aldrich).

**Quantification of α-amylase in saliva using enzymatic activity assay**. The concentration of α-amylase in saliva was analyzed using standard protocol established in Queensland Health Chemical pathology laboratories for Beckman AU480 clinical chemistry analyzer that utilize the amylase assay kit of the manufacture (IFCC G7-PNP, catalogue number OSR6006).

**Reporting summary**. Further information on research design is available in the Nature Research Reporting Summary linked to this article.

## Data availability

The authors declare that the data supporting the findings of this study are available in this paper and its supplementary information files. Source data are provided with this paper. All mass spectrometry raw data files are available via ProteomeXchange with identifier PXD031169. Source data are provided with this paper.

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

## Acknowledgements

We thank Dr. Carel Pretorius from Pathology Queensland for carrying out quantification of biomarkers in human samples for this study. This work was supported in part by the Australian Research Council Discovery Projects DP160100973 and DP150100936, as well as ITTC grant IC160100027 and NHMRC grant APP1113262 to K.A. This work was also in part supported by HFSP grant RGP0002/2018 to K.A. This work was supported in part by the National Institute of General Medical Sciences of the National Institutes of Health under Grants R01GM118675 and R01GM129291 to H.A. K.A. gratefully acknowledges financial support of QUT/CSIRO Synthetic Biology alliance. J.J.P. is supported by a UKRI Future Leaders Fellowship (MR/T02223X/1) and is a Turing Fellow.

## Author contributions

Z.G. designed experiments, performed experiments, analysed data and wrote manuscript, R.D.P. designed experiments, performed experiments, analysed data, Y.X. designed experiments, provided reagents, analyzed data, W.A.J. designed experiments, performed experiments, analysed data, and wrote manuscript, P.W. performed experiments and wrote the manuscript, S.E. designed experiments, performed experiments, analysed data and wrote manuscript, S.V.M. designed experiments, performed experiments, J.P.J.U. designed experiments, provided samples and analysed data, H.A. designed experiments, performed experiments, analysed data, provided reagents and wrote manuscript, J.N. designed experiments, performed experiments, analysed data, wrote the manuscript, K.A. designed experiments, analysed data and wrote manuscript. The manuscript was written through contributions of all authors. All authors have given approval to the final version of the manuscript.

## Competing interests

The authors declare no competing interests.
