## [Peer Review File · Nature Communications]

Reviewers' Comments:

Reviewer #1:

Remarks to the Author:

Here the authors build synthetic, single-component, allosterically ligand-gated versions of NanoLuciferase (Nanoluc). They circularly permutate NanoLuc by connecting the existing N and C termini with a long linker, resulting in new N and C termini somewhere in a loop connecting two beta-strands in luciferase. Then they fuse different ligand-binding proteins to these new termini to build small protein sensing sensors. Despite lack of novelty, I find this is a well-conducted and work that is necessary to advance the biosensor field, so I support its publication in Nature Communications if they can address a few points I describe below. With enthusiasm, I particularly like the hydrogen-deuterium exchange experiments to explain the structural mechanism of sensors. Here are my comments and questions:

1. This point is my major concern. I think a clearer goal and experiments to support this goal would make this study more complete. For example, if the idea is to make better sensors to detect analytes with more sensitivity, then the work lacks the experiments to quantify the analytes directly from blood and plasma samples. If the goal is to make better sensors to address a biological question that is not possible with the current sensors, then the work lacks in-cell or animal studies. I think one of these directions with the supporting data would make this work more significant and impactful.
2. In the first experiments, the authors fuse FKBP and FRB to the new termini and purify the recombinant protein from E.coli and test rapamycin dependent luminescence. If the sensor protein is structurally disordered/unfolded, how did the authors achieve to purify the disordered recombinant protein without rapamycin, which usually tends to form inclusion bodies?
3. In Figure 2, the plots are shown as relative luminescence on the y-axis. Is there any background luminescence without the ligand? Also, what are the Hill coefficients for these titration curves? The Hill coefficients would tell the cooperativity between ligand (rapamycin, tacrolimus, amylase) binding and luciferase substrate or luciferase activity? These would then help to support the structural mechanisms derived from HD/X assays coming later in the paper.
4. In Figure 4, it is a bit hard to read the figures. I think it would be better if they make the table smaller and enlarge the plots (panel A). It would also clarify if the rapamycin binding sites on the diagram (panel A). Also, why is the signal on cpNanoLuc 30s plot disappear on 300s plot and then increase in 3000s plot? Also, rapamycin affinity is known to be sub-nanomolar for FKBP and FRB. Why is the range around 2-15- μ M?
5. Along with the previous point, are there intermolecular interactions in the presence of rapamycin? For example, does FKBP of one sensor molecule interact with FRB of another molecule in the presence of rapamycin? Can we understand (or explain high rapamycin concentration-dependent structural changes) whether this is the case from the plots on Fig. 4A?
6. How about the reversibility of binding? If the analyte is removed, how is the response of sensors change? This feature would be particularly useful for in-cell studies.
7. In the motivation of the work, the authors claim that no general principle has been established to build synthetic switches. I suggest the authors take a look at these papers which have similar ideas and approaches.

Dagliyan et al. Engineering extrinsic disorder to control protein activity in living cells. Science.

Dagliyan et al. Rational design of a ligand-controlled protein conformational switch. PNAS

Dagliyan et al. Engineering Pak1 allosteric switches. ACS Synthetic Biology

Dagliyan et al. Computational design of chemogenetic and optogenetic split proteins. Nature Communications

Dagliyan et al. Controlling protein conformation with light. Current Opinion in Biotechnology

Dagliyan et al. Engineering proteins for allosteric control by light or ligands. Nature Protocols

Reviewer #2:

Remarks to the Author:

This work represents a solid follow-up to the same group's 2019 JACS paper where they now describe improved small molecule sensors based on NanoLuc luciferase.

Experiments appear to be performed well and in a way the data can be trusted and informative. But the paper doesn't provide much that is extraordinarily new or significant in terms of developing enabling sensors that are generalizable.

The major concern is one of messaging. The authors refer to their achievements as evidence of a technology with potential for general applicability to making sensors for any small molecule. Most of their evidence comes from a very specific system: rapamycin inducible complex formation between FRB and FKBP, where the presence of rapamycin in a sample drives signal generation by inducing the complexation of FRB and FKBP. Though the data indicates the circularly permuted NanoLuc works well for this, the assay is specific to the detection of a molecule requiring two proteins that are capable of forming a complex. This implies that for general applicability a small molecule of interest must have the capability of bringing two proteins together. Is this common enough to conclude the method is generalizable?

Remaining comments/critiques are more technical in nature or minor...

1. The luciferase should be referred to as NanoLuc luciferase or in abbreviated forms as either NanoLuc or NLuc. This will give consistency with most literature, including the original description of the enzyme development by Promega.
2. The split NanoLuc work described by Dixon et al (ref#19) indicates that the binary NanoLuc system was discovered using a small library of circularly permuted sequences. The same dissection point was used in the authors' 2019 JACS paper as well as this submitted work. Dixon et al did not add additional sequence to the new termini. I would be curious to know if adding extra sequence would have made any difference. I realize this is a bit out of scope but those pieces can complement in sometimes very orientation dependent manners. I'm wondering if it's sterics or energy, or entropy playing a role. Perhaps all. Lastly, and most relevant to this line of questions... Did the authors test two-piece system like in the 2019 JACS paper but with the last B-strand as the N-terminus (i.e., N-NLuc1-161--CAM--B10--FKBP-C + N-FRB--CAMbp-C).
3. Did the authors consider modifying the sequence of the B10 peptide in their constructs? For modifying affinity? Would it be worth trying?
4. In the end how valuable is the energy/bonding data? How will data from the rapamycin system provide useful information for other types of sensors?

Reviewer #3:

Remarks to the Author:

This study develops a molecular biosensor using a circularly permuted nanoLuciferase construct linked to ligand-binding species (e.g., FKBP/Rapamycin/FRB). The concept of the study is interesting and scientifically sound, but the authors substantially overstate the relevance of the work, in my view, do not explain a significant fraction of their results in a way that I can understand (including in areas that I really should understand given my background), and offer a general quality of writing that is below what I would expect for submission to Nature Communications. The work would most likely be suitable for publication in a mid-tier bioanalytical journal(after major revisions) or one that does not consider prospective impact, but is not suitable for publication in Nature Comms, in my view. Specific comments are below:

1. Overstating importance: This issue starts with the hypothesis - "Here we hypothesize that

circular permutation of proteins increases the probability of functional coupling of new N- and C- terminal sequences with the active center of the protein through increased local structural disorder". This hypothesis may sound general, but it is actually specific to circular permutation of proteins like nanoLuc in which the N- and C- termini interact (probably also needs to be in a beta sheet, also like nanoLuc). The other idea, that disordered regions can be re-ordered through ligand binding to generate a signal is also far from novel. There are innumerable molecular probes - including a host of complementation assay probes that operate on essentially the same principle. Though this lens, the paper introduces a new order-based probe system with apparently (good analytical properties. That's not insignificant, but does not rise to the level of the kind of advancement normally reported in a top tier journal.

2. In many cases, I find it difficult to understand what the authors have actually done, particularly in the reporting of analytical results. The HDX data are a particularly vexing example of this - either the data are not clearly reported or the authors don't understand them at all. Some of the questions that come to mind: If "HDX difference" is being reported, what does the "0um Rapamycin" trace mean? I suspect this may be straight 'uptake', but if it is, why does it go *down* as a function of time in some regions (particularly evident in the 0um Rapamycin trace). How did the authors achieve single amino acid resolution? In the manuscript, it is left to appear that the measurements were for single amino acids (which is almost always not possible using the conventional workflow employed here), but in the methods a vague description is given of 'linearly weighting peptide level measurements'. It is hard to know what this means, exactly, but what it sounds like would not be appropriate, since HDX uptake at each site is essentially a *unique* pseudo-first order process. The authors could and should have simply reported the uptake kinetics for the peptides they were actually measuring that contained the desired amino acid(s). The paper they cite for this [28] sounds like it would be relevant to single amino acid resolution, but actually really isn't.

Ultimately, I would recommend that the authors do better justice to the many existing molecular biosensors that rely on ligand-induced order changes in the introduction and discussion (this is currently mentioned in the introduction, but needs a more thorough exploration with direct comparison to the current sensor in the discussion). The discussion of analytical results could also be substantially improved, especially in the HDX section. A much simpler presentation for Figure 5 in particular, with large panels for 5A, would in itself be a substantial improvement.

We addressed the specific criticism of the reviewers in the following way:

Reviewer #1

Point 1: This point is my major concern. I think a clearer goal and experiments to support this goal would make this study more complete. For example, if the idea is to make better sensors to detect analytes with more sensitivity, then the work lacks the experiments to quantify the analytes directly from blood and plasma samples. If the goal is to make better sensors to address a biological question that is not possible with the current sensors, then the work lacks in-cell or animal studies. I think one of these directions with the supporting data would make this work more significant and impactful.

Response: Primarily this is a protein design paper where we explore the entropically driven OFF states as an approach for constructing integrated artificial allosteric systems. These designs get around the need for ligand binding domains with large conformational changes that are in a short supply. We have significantly modified the introduction and discussion sections to make those points. We also now refer to the concept of non-covalent intramolecular cyclisation as another way of looking at the ligand-mediated reporter domain ordering event. One of the benefits of such design is in the ability of the developed biosensors to detect to both to small molecule and protein biomarkers. This sets the approach apart from all previously developed integrated protein biosensor platforms- the point that we now brought further to the foreground in the discussion section.

To further evidence the practical utility of the developed luminescent biosensors we now present new data where these biosensors were used to quantify biomarkers such as human α -amylase and human serum albumin in saliva and serum respectively. These data are now presented as Figures 3 A,B and Figure S6. We also present the evidence that the BRET biosensors with red emission can be used to quantify analytes in blood (Fig. 3E,F). Taken together this data clearly supports the practical utility of the developed biosensor platform. We also extended the discussion section to point out that unlike all other single biosensor systems the presented platform can be applied both to small molecules and protein biomarkers alike.

Point 2: In the first experiments, the authors fuse FKBP and FRB to the new termini and purify the recombinant protein from E.coli and test rapamycin dependent luminescence. If the sensor protein is structurally disordered/unfolded, how did the authors achieve to purify the disordered recombinant protein without rapamycin, which usually tends to form inclusion bodies?

Response: This is a very important question as it affects our ability to make biosensors predictably.

To further analyse folding and oligomerisation state of the cpNanoLuc-based biosensors, we now performed size exclusion chromatography experiments comparing the elution profiles of the cpNanoLuc rapamycin biosensor in the presence or absence of rapamycin. The profiles were indistinguishable regardless of the ligand presence and corresponded to a monomeric form of the biosensors. This observation is in line with the idea that the developed molecule is largely structured, and its activity controlled by localised folding events. These data are now presented as Figure S1D and discussed in the main text.

Furthermore, the HDX-MS data shows that disordering of cpNanoLuc is relative subtle and affects only a part of the molecule. Certain level of disorder that may lead to a dramatic decrease of protein activity and yet may not be significantly destabilising for overall fold. This, for instance, is well documented in, now referenced studies, that utilise domains insertions to create artificial OFF states. A related observation it that it is well established that NanoLuc can be produced in folded form when lacking its last β -strand – a phenomenon that has been exploited to create Promega's NanoBiT protein:protein interaction detection system (Dixon et al. 2016).

Point 3. In Figure 2, the plots are shown as relative luminescence on the y-axis. Is there any background luminescence without the ligand? Also, what are the Hill coefficients for these titration curves? The Hill coefficients would tell the cooperativity between ligand (rapamycin, tacrolimus, amylase) binding and luciferase substrate or luciferase activity? These would then help to support the structural mechanisms derived from HD/X assays coming later in the paper.

To address the first question of the reviewer we now present the raw ligand titration data in as part of the plot in the figure S1A-C for cpNanoLuc-based biosensors and in the figure 6C,D for cpLumiLuc-based biosensors. On request of the referee, we now show the Hill co-efficient plots for the titrations from Figure 2 as Fig. S1. The fit of the data leads to Hill co-efficient of slightly above 1 that may indicate some positive cooperativity between the ligand event and NanoLuc's activity in rapamycin and α -amylase biosensors. However, such information may be difficult to extract from these experiments with confidence as the NanoLuc substrate is present in much higher concentration (mid μ M) than the biosensor (1 nM) changes in NanoLuc's affinity for the substrate may not be detectable in the assay. Furthermore, the

ligand binding and isomerisation events are expected to operate on very different scales (milliseconds and minutes respectively) and their putative interactions may not be resolvable well in endpoint enzymatic assay.

4. In Figure 4, it is a bit hard to read the figures. I think it would be better if they make the table smaller and enlarge the plots (panel A). It would also clarify if the rapamycin binding sites on the diagram (panel A). Also, why is the signal on cpNanoLuc 30s plot disappear on 300s plot and then increase in 3000s plot? Also, rapamycin affinity is known to be sub-nanomolar for FKBP and FRB. Why is the range around 2-15- μ M?

We have significantly restructured the presentation of these results in Fig. 4 and the accompanying text in an effort to make them more easily understood. The table has been extracted from Fig. 4 to a separate table. The deuterium labeling (panel A) is presented with a consistent y-axis, mark-up of the biosensor protein domains on the x-axis and clearer color coding of the rapamycin titration. The figure legend now explicitly states that 8 mM biosensor was used in these experiments, which directly relates to why micromolar concentrations of rapamycin were used in order to generate stoichiometric ratio of ligand-bound biosensor. Also the following statement was added to the text describing the HDX-MS results: "The rapamycin biosensor was at 8 μ M for this experiment and so the rapamycin was added between 0-20 μ M, i.e. up to a 2.5:1 stoichiometry. This ensured binding saturation, given the Kd value estimate of 0.5 nM from a luminescence assay."

5. Along with the previous point, are there intermolecular interactions in the presence of rapamycin? For example, does FKBP of one sensor molecule interact with FRB of another molecule in the presence of rapamycin? Can we understand (or explain high rapamycin concentration-dependent structural changes) whether this is the case from the plots on Fig. 4A?

It is possible for stable intermolecular (rapamycin-mediated non-covalent crosslinking) interactions to occur, although these would be difficult to distinguish by HDX-MS data. As the effective local concentration of each intramolecular FKBP and FRB domain is very high, since they cannot diffuse away from each other, the k_{on} is anticipated to be fast. So, fractional population of a dimer or higher oligomer species should be low. HDX-MS data will be derived from an ensemble average and the measurements are anticipated to be dominated by the monomeric species.

6. How about the reversibility of binding? If the analyte is removed, how is the response of sensors change? This feature would be particularly useful for in-cell studies.

This is an a very important point that, due to very high affinity of the developed biosensors for their ligands, we struggled to address in the original submission. We now developed an assay based on the surface-immobilised cpNanoLuc HSA biosensor that enabled us to address this point. Using this assay, we were able to wash away large fraction of bound HSA and revert the activity of the biosensor. We were then able to re-activate the biosensor by re-adding HSA. This data is now presented in the figure S4 and discussed in the main text on page 8 confirming that the developed biosensor architecture is reversible.

Point 7: In the motivation of the work, the authors claim that no general principle has been established to build synthetic switches. I suggest the authors take a look at these papers which have similar ideas and approaches.

Response: We thank the reviewer for rising an important question about the level of the design abstraction and the differentiation from the prior engineering efforts. The reviewer points to a range of pioneering studies on construction of ligand or light regulated proteins. These are based on synthetic OFF states created by domain insertion that is then relieved by a ligand-mediated structural re-arrangement. These approaches are related to the one we present here as all of them rely on creation of disorder-mediated OFF state. The major difference is that in the presented design the OFF state is created by , presumably, entropic forces and does not utilise an exogenous regulatory domain. The reversal of the disorder is achieved by, essentially, by non-covalent cyclisation event that reduces the disorder though overall constraint and, as a result , increases the catalytic activity of the system. This may help to explain why the system functions similarly when using cooperative binders of small molecules or non-cooperative binders of protein ligands. This also explains why binders with very different size and topology can be utilised to construct cpNanoLuc -based biosensors.

We now have re-written the introduction section to provide a more global view on approaches for construction of reversible OFF states in proteins and significantly increased the number of cited publications. We also modified the discussion section and labour on the molecular mechanisms of the developed biosensors and its differentiation from other integrated protein switches.

Reviewer #2

Point 1: The major concern is one of messaging. The authors refer to their achievements as evidence of a technology with potential for general applicability to making sensors for any small molecule. Most of their evidence comes from a very specific system: rapamycin inducible complex formation between FRB and FKBP, where the presence of rapamycin in a sample drives signal generation by inducing the complexation of FRB and FKBP. Though the data indicates the circularly permuted NanoLuc works well for this, the assay is specific to the detection of a molecule requiring two proteins that are capable of forming a complex. This implies that for general applicability a small molecule of interest must have the capability of bringing two proteins together. Is this common enough to conclude the method is generalizable?

Response: The reviewer raises a very important point of the general applicability of the developed system. As indicated above in the response to the Point 7 of the referee 1 we significantly modified the introduction and discussion sections to better frame the scientific question and the obtained results. The main differentiation of the system from other biosensor architectures is in the, presumably, entropically driven OFF state that can be counteracted by the ligand-mediated non-covalent intramolecular cyclisation. As result the system is tolerant to the topology of the binding entities as demonstrated by α -amylase and human serum albumin biosensors as well as biosensors of tacrolimus and rapamycin. Development of protein binding domains is quite straightforward, and we demonstrate that commonly used synthetic binders such as VHH domains are suitable for this purpose. Moreover, a significant progress has been made in bottom-up construction of artificial systems where two proteins associate in a small molecule-dependant fashion (also known as chemically induced dimerization systems). We now expanded the discussion of these advancements and provided additional references related to the progress in the methodologies for the development of the small molecule binders and chemically induced dimerization systems.

Point 2: The luciferase should be referred to as NanoLuc luciferase or in abbreviated forms as either NanoLuc or NLuc. This will give consistency with most literature, including the original description of the enzyme development by Promega.

Response: Corrected as suggested

Point 3: The split NanoLuc work described by Dixon et al (ref#19) indicates that the binary NanoLuc system was discovered using a small library of circularly permuted sequences. The same dissection point was used in the authors' 2019 JACS paper as well as this submitted work. Dixon et al did not add additional sequence to the new termini. I would be curious to know if adding extra sequence would have made any difference. I realize this is a bit out of scope but those pieces can complement in sometimes very orientation dependent manners.

Response: This is an important point that we unfortunately have not analysed in the detail in our paper. However, we discuss in the opening paragraph of the results the idea that newly designed termini in circularly permuted proteins are more prone to spontaneous dissociation and disorder as they have not undergone evolutionary selection against those features. This idea cannot be really tested by a single design and will require analysis of a much larger set of circularly permuted proteins. However, HDX-MS is a method that could provide answers on the level of structural disorder in such library. We discuss this point now at the end of the paper.

Point 4: I'm wondering if it's sterics or energy, or entropy playing a role. Perhaps all.

Response: Our data does not allow to resolve the contribution of these possible factors individually but we speculate in the discussion section that the entropic and solvation forces play the most significant role in this process. We have added the following statement to the text for the HDX-MS analysis section: "The unbound FKBP domain exchanges on average twice as much deuterium label as the FRB domain (average fractional uptake for all peptides at all time points 0.29 compared to 0.14). However, the average predicted intrinsic HDX rates are similar (1.8 compared to 2.0 s⁻¹), solvent accessible surface area (SASA) for the domains is similar (6086 compared to 6320 Å²) and the surface burial by rapamycin is similar (532 compared to 736 Å²). This indicates that there is a far greater stabilization of the FKBP domain upon rapamycin, likely with a large entropic component."

Point 5: Lastly, and most relevant to this line of questions... Did the authors test two-piece system like in the 2019 JACS paper but with the last B-strand as the N-terminus (i.e., N-NLuc1-161--CAM--B10--FKBP-C + N-FRB--CAMbp-C).

Response: We have not generated this chimeric protein but from our experience about 10% of calmodulin insertions into a host protein will produce OFF to ON switch (never the reverse). The switches differ in the dynamic range and the response time but otherwise quite similar in their behaviour. We expect that additional switchable CaM-NanoLuc chimeras can be constructed from both circular permuted and native NanoLuc sequence. We, however, had the objective of developing an approach that would not rely on the domains insertion and therefore sought the way to create local structural disorder through a different mechanism. We now brought this notion to the foreground both in introduction and discussion section.

Point 6: 3. Did the authors consider modifying the sequence of the B10 peptide in their constructs? For modifying affinity? Would it be worth trying?

Response: We have not performed these experiments, but they are feasible and are likely to lead to changes of dynamic range and system's response time. An interesting question if there is a balance between the size/structure of the binding domain and the sequence of the terminal peptides. Assuming that larger domains will be subjected larger solvation forces one can expect that they will exert a larger pulling force on the peptide increasing the dominance of the OFF state. We now expanded the discussion section to capture those ideas.

Point 7: In the end how valuable is the energy/bonding data? How will data from the rapamycin system provide useful information for other types of sensors?

Response: As all four biosensors constructed in the present study behaved very similarly and could be converted to LumiLuc and BRET biosensors. They also all demonstrated a significant sensitivity gain when compared to the two component biosensors based on the same domains strongly suggesting that they utilise the same mechanism. Therefore, we feel that general conclusions about the mechanistic aspects of the biosensors function are warranted but point to the avenues of further investigation in the discussion section. Furthermore, we demonstrate here the potential for HDX-MS to pinpoint the local regions of a protein that are altered upon ligand binding, but which are least sensitive to the ligand and would therefore be excellent targets for mutation in rational engineering of higher affinity variants.

Reviewer #3

Point 1: Overstating importance: This issue starts with the hypothesis - "Here we hypothesize that circular permutation of proteins increases the probability of functional coupling of new N- and C- terminal sequences with the active center of the protein through increased local structural disorder". This hypothesis may sound general, but it is actually specific to circular permutation of proteins like nanoLuc in which the N- and C- termini interact (probably also needs to be in a beta sheet, also like nanoLuc). The other idea, that disordered regions can be re-ordered through ligand binding to generate a signal is also far from novel. There are innumerable molecular probes - including a host of complementation assay probes that operate on essentially the same principle. Though this lens, the paper introduces a new order-based probe system with apparently (good analytical properties. That's not insignificant, but does not rise to the level of the kind of advancement normally reported in a top tier journal.

Response: Some of the review's concerns overlap with those of reviewer 1 and dealt in detail in response 7 above. To that we can add that there is a large difference in performance and utility of two component and fully integrated biosensors that it discussed in the introduction of the paper. The dependence of the output on the biosensor component's concentration is one of the central problems of the multi-component biosensors. With exception of small molecule semi-synthetic dissociative BRET biosensors such LUCID introduced by Johnsson and colleagues single component biosensors require domains with significant conformational changes upon ligand binding. None of the integrated single component biosensors are generally applicable and are typically utilised for detection of small molecules. The presented architecture address these limitations in the ways discussed above and in the modified version of the manuscript.

*Point 2. In many cases, I find it difficult to understand what the authors have actually done, particularly in the reporting of analytical results. The HDX data are a particularly vexing example of this - either the data are not clearly reported or the authors don't understand them at all. Some of the questions that come to mind: If "HDX difference" is being reported, what does the "0um Rapamycin" trace mean? I suspect this may be straight 'uptake', but if it is, why does it go *down* as a function of time in some regions (particularly evident in the 0um Rapamycin trace). How did the authors achieve single amino acid resolution? In the*

*manuscript, it is left to appear that the measurements were for single amino acids (which is almost always not possible using the conventional workflow employed here), but in the methods a vague description is given of 'linearly weighting peptide level measurements'. It is hard to know what this means, exactly, but what it sounds like would not be appropriate, since HDX uptake at each site is essentially a *unique* pseudo-first order process. The authors could and should have simply reported the uptake kinetics for the peptides they were actually measuring that contained the desired amino acid(s). The paper they cite for this [28] sounds like it would be relevant to single amino acid resolution, but actually really isn't.*

We have made very significant changes to the presentation and discussion of the HDX-MS results in light of the above concerns. In particular, the deuterium uptake data for individual peptides as a function of rapamycin concentration (Figure 4A) are more clearly presented and annotated. A complete set of uptake plots is provided in accompanying documents (with the x-axis automatically labelled by the HDX software, DynamX, as "Exposure Time" whereas it is actually rapamycin concentration in micromolar). Each panel in Figure 4A has the same y-axis for ease of comparison and this is shown as "fractional deuterium uptake". The x-axis is annotated with the biosensor protein domains and glycine-rich linkers. The accompanying text now more clearly explains the methodological approach that we used and straightforwardly describes the data treatment. The rationale for the experiment is more thoroughly explained in the text, including explicit comment on the biosensor concentration (8 μM) and thus the range of rapamycin concentrations used to achieve stoichiometric complex. The unprocessed peptide-level data is presented in Figure 4B with examples from each cluster to evidence the dose-response relationship that is present in the data, even before flattening it to the amino acid level. We have cited the relevant reference for Keppel and Weis' method of processing peptide-resolution HDX-MS data to amino acid-resolution: Keppel, T. R., & Weis, D. D. (2015). Mapping residual structure in intrinsically disordered proteins at residue resolution using millisecond hydrogen/deuterium exchange and residue averaging. *Journal of the American Society for Mass Spectrometry*, 26(4), 547–554. <https://doi.org/10.1007/s13361-014-1033-6>. We have attempted to more clearly show that there is coherent behavior of distinct sub-molecular regions of the biosensor with the inclusion of Figure panels 4C-D and the extraction of the table into a separate item.

Point 3. Ultimately, I would recommend that the authors do better justice to the many existing molecular biosensors that rely on ligand-induced order changes in the introduction and

discussion (this is currently mentioned in the introduction, but needs a more thorough exploration with direct comparison to the current sensor in the discussion). The discussion of analytical results could also be substantially improved, especially in the HDX section. A much simpler presentation for Figure 5 in particular, with large panels for 5A, would in itself be a substantial improvement.

As mentioned above we significantly modified the manuscript to better frame the experimental data and conceptual differentiation of the presented designs.

Reviewers' Comments:

Reviewer #1:

Remarks to the Author:

I think the authors have addressed my questions, so I support its publication in Nature Communications.

As a minor suggestion, the authors should triple-check the panel labels (for. ex. missing in Figure 3), axis labels in graphs, and figure citations in the text.

Reviewer #2:

Remarks to the Author:

From my previous point #1: It's better now, and there is now more evidence of modularity. But I would still be careful not to overplay this argument. Perhaps putting it into context of what David Baker's group has recently done with their lock and key system using proteins from scratch would be relevant and informative to the reader (as a general strategy but also in terms of modularity).

Prev pt #3: It looks like they still need to cite Dixon et al 2016 for the sequence they use for their sensor. Dixon et al built something very similar and essentially used it as a protease sensor (though indirectly). Not surprising the authors missed this as it's mainly found in the supplementary info.

I'm still curious if a simple experiment could be done where the ends of a reporter (or a cp reporter) could be extended artificially with flexible amino acids unlikely to add significant structure (e.g., a series of Ser/Gly repeats perhaps). It continues to feel like this sort of contrived experimental design could be useful for the arguments on mechanism being attempted in this report.

Prev pt #5: The response provided for this point is appreciated and helpful. It should be added to the actual discussion in the manuscript as it really highlights the utility of the asymmetric split in Nluc providing the b10 peptide with a flexible sequence that should be able to be dialed in for different affinity needs.

1. The report feels too heavy with theoretical speculation on models for protein allostery and modularity. It's interesting discussion don't get me wrong, and belongs in the paper. However, it should be restricted to the discussion section so that it is more clear it is speculation. Plus, this approach would give more opportunity to speak to the result-driven evidence for how well the current sensors work and where their future utility lies.
2. Did the authors try a non-permuted form of Nluc for their sensors? Curious if fusing the rapamycin (or similar) system to the native termini could work. If it failed or worked but not as well as the cp version this would give credence to the authors case for building the technology using a cp enzyme.
3. In "Construction of permuted NanoLuc luciferase" section of results it is implied that Renilla luciferase is ATP-dependent when it actually is not. Also in this section, NanoLuc luciferase is referred to as 'an engineered version of NanoLuc luciferase'. This doesn't make sense. It is assumed the authors mean to say that NanoLuc luciferase is an engineered enzyme derived from the native luciferase from *Oplophorus*. This should be updated to be more technically accurate.
4. Figure 1 is not clear. It is hard to understand the various ribbon structures. Termini are difficult to see. Better color coating could be used to know where strands 9, 10 are in panel C. The unstructured sequences could be labeled as strands 9, 10 to help with this. The authors could also consider adding a cartoon similar to what Dixon et al. used to describe the architecture of the split NanoLuc system (i.e., where strands 9, 10 are located).
5. For the cp NanoLuc how is this sequence different from what Dixon et al published in 2016. They made a similar cp NanoLuc with a flexible linker connecting the former termini. If the sequence is as similar as it looks this could be made more clear and Dixon et al. should be cited.
6. For the cp NanoLuc what is the linker between former termini? Is it important to have a certain

type of linker (e.g., hydrophobic, hydrophilic, flexible, rigid) or a precise linker length? Were these variables examined? If not could the system be further improved by targeting this region of the sequence?

7. For Figure 2 what is Calcineurin A and Calcineurin B? It feels like it is overly assumed that the reader will know what these are and how they relate to the rap system. Some elaboration would be helpful.

8. In figure 2 what are the raw RLU values? Could this be mentioned or added to supporting info? How does the luminescence intensity of the sensor compare to unmodified NanoLuc?

9. VHH domains refer to Nanobodies; is that correct? If so this could be made more clear. On a related note could the authors comment on how dependent their technology is on the existence of nanobodies or other binding moieties.

10. Under the section "Functional mechanism of cpNanoLuc biosensors" what is the "activation delay" that the authors refer to? What is the magnitude of the delay and what is the perceived mechanism/explanation? How big a detriment is the delay?

11. The authors state their data indicates a greater stabilization of FKBP than FRB upon binding to rap. What can be said about the impact of rap binding on Nluc structure (independent of inducing proximity between strands 9, 10)?

12. Which strand are the authors referring to when they say that b-2 of the cpNanoLuc is detached from the b-sheet. Which strand is that? The original strand 1? And which b-sheet are they referring to, the original 10 that's now the N-term?

13. Figure 5 is very difficult to follow because there is such poor resolution with the structure images. Cartoons or even some form of text descriptions would be easier to follow. I suppose the figure could stand as is if moved to the supporting info.

Reviewer #3:

Remarks to the Author:

This is a resubmission of a manuscript on an engineered NanoLuc based sensor. One of my main concerns for the last version of this manuscript was that the authors were making over-general claims about this switch stating what I felt were overstated claims of novelty and impact. This view was, I think, shared by one of the other reviewers. The other main concern was the HDX data, which I found difficult to comprehend in any sense.

Overstated claims: The authors have toned this down somewhat and do a somewhat better job of focusing on what is my view the most novel aspect, which is the ability to accommodate a wide range of 'binding domains' in this switch. Nonetheless, the advance reported here is still considerably less novel, in my opinion, than the authors are indicating, especially given their recent work on similar sensors. If the authors were genuinely setting out to prove their general hypothesis that circular permutants will often be suitable for this type of sensor design, for example, then they would have had to generate several sensors from other circularly permuted proteins.

HDX data: The authors have also improved this section substantially, but now conduct a highly unusual (and I think misguided) analysis of the data. The authors appear to be operating under the assumption that the midpoint of peptide-specific titration curves reflect the 'sensitivity' of certain regions to the presence of ligand. In fact, (assuming that k_{on} is large relative to the chemical exchange rate and $1/k_{off}$ is large relative to the labeling time, which is likely the case given the low K_d) there would be, at sub-stoichiometric concentrations of ligand, two uptakes for each (affected) peptide, one for the bound state and one for the unbound state, corresponding to distinct distributions with amplitudes linked to their relative populations. When these distributions have the same amplitude, (which should occur at just over 4 μM ligand for *every peptide*) then the ligand is 50% bound. Differences in the extent to which a given region is structurally impacted by ligand binding (i.e., the difference in uptake between the bound and unbound states) would impact the magnitude of the sigmoid, but not its mid-point, which would always be when the ligand is 50% bound... So why did the authors measure different EC_{50} s on different peptides? Possibly because of confounding factors like the rate at which conformational equilibrium is established in particular regions after the ligand binds (k_{on}) or unbinds (k_{off}) or non specific interactions with ligand (this second explanation is presented by the authors as a reason why most

of their EC₅₀ values don't make sense, given the low k_d of the ligand, which is a reasonable explanation, but it is another inexplicable thing about the data).

I do think that the time-dependent changes in HDX uptake at stoichiometric+ concentrations of ligand more-or-less confirm that ligand binding induces restructuring of the protein, particularly in the N and C-terminal regions, and it would be far preferable, in my view, if the authors simply conducted this more conventional analysis.

We addressed the specific criticism of the reviewers in the following way:

Reviewer #1

Point 1: As a minor suggestion, the authors should triple-check the panel labels (for. ex. missing in Figure 3), axis labels in graphs, and figure citations in the text.

Response: The repetitive axis labels that were originally omitted in some panels are now restored. We carefully checked the text for accuracy of figure citations.

Reviewer #2

Point 1: From my previous point #1: It's better now, and there is now more evidence of modularity. But I would still be careful not to overplay this argument. Perhaps putting it into context of what David Baker's group has recently done with their lock and key system using proteins from scratch would be relevant and informative to the reader (as a general strategy but also in terms of modularity).

Response: This is a very good point as during the revision period of our manuscript Merxx's, Wells's and Baker's groups presented different versions of two component NanoLuc-based biosensor systems. The Baker's design is the most original and technically sophisticated and is reliant on the proximity driven concentration changes of the components controlled by steric hindrance of the analyte:biosensor complex. While this is a remarkable protein engineering effort it is still yet another two-component NanoLuc architecture and, for instance, figure 2D of our study and associated reference 14 depict an alternative two component NanoLuc biosensor system developed by us. The use of the cage component in Baker's design enables reduction of the background and, presumably, extends the concentration range at which such biosensor could operate. However, the idea of using a caged activator to achieve those advantages was published by us previously (ref 21) and such caged activators are compatible with the two component NanoLuc biosensor shown in figure 2D. Furthermore, the reliance on steric hindrance of Baker's architecture limits its applicability to protein analytes- a limitation that our NanoLuc biosensors do not have. Importantly, fully integrated single component biosensors have several significant differentiating features:

a) They are fundamentally not susceptible to "hook" effect – where oversaturation with the analyte leads to progressive decrease of signal.

- b) They are more sensitive due to the presence of the dimerization system in *cis*-
- c) they are not limited to a particular analyte class.
- d) They are more suitable for in vivo applications than multicomponent biosensors that require co-delivery and dose/ratio optimisation of the components.

In order to avoid turning the discussion section of the manuscript into a review article we added the following sentence on the page 17 of the revised manuscript version:

While a range of two component NanoLuc biosensors of various designs and complexity have been reported, to our knowledge this is the first single component biosensor architecture that can be targeted to such structurally diverse analytes^{14,37;27;47;48;49,50}.

Point 2: Prev pt #3: It looks like they still need to cite Dixon et al 2016 for the sequence they use for their sensor. Dixon et al built something very similar and essentially used it as a protease sensor (though indirectly). Not surprising the authors missed this as it's mainly found in the supplementary info.

Response: We thank the reviewer for pointing us to the data in the supplementary section of the Dixon's paper that we have, regretfully, overlooked. This is indeed a very similar circular permutation of the NanoLuc that allowed the authors to create a protease-controlled OFF switch. To reflect this pre-existing design, we now introduced the following sentence on the page 5:

*A similar circular permutation of NanoLuc was previously constructed where cleavage of the linker connecting β -strand 10 with 1 by a protease led to the loss of activity*²⁸.

Point 3: I'm still curious if a simple experiment could be done where the ends of a reporter (or a cp reporter) could be extended artificially with flexible amino acids unlikely to add significant structure (e.g., a series of Ser/Gly repeats perhaps). It continues to feel like this sort of contrived experimental design could be useful for the arguments on mechanism being attempted in this report.

Response: We thank the referee for the suggested experiment that turned out to be quite informative. The experimental data is now included as figure S1E. The difference between

the constructs with N and C- termini of different length turned out to be smaller than we expected, indicating that the performance of our biosensor is predominately governed by the structure its core component and less by the structure of the receptor domains. This is in fact, in line with the observation that the use of a broad range of binding domains of different sizes and structures resulted in biosensors with similar background activities and dynamic ranges. To reflect these conclusions, we introduced a following sentence on the page 6. See also the answer to point 6 below:

We further compared the background activity of cpNanoLuc with its variant containing 30 glycine/serine repeats on both N and C termini (Table S1). The mutants with extended termini had similar background activity than its parental variant indicating that the size terminal extensions is less important than we initially anticipated (Fig. S1E). However, this is a useful feature that potentially enables utilization of structurally diverse ligand binding domains for biosensor construction.

Point 4: Prev pt #5: The response provided for this point is appreciated and helpful. It should be added to the actual discussion in the manuscript as it really highlights the utility of the asymmetric split in Nluc providing the b10 peptide with a flexible sequence that should be able to be dialed in for different affinity needs.

Response: On request of the referee, we now explicitly mention the possibility of tuning the performance of the biosensors by modifying the sequence of β -strands 10 on the top of the page 18 where we introduce the following phrase:

In particular, we expect that by modifying sequences of β -strands 1 and 10, as well as the length and composition of the linkers connecting the cpNanoLuc and binding domains it will be possible to modulate the dynamic range and response times of the system.

Point 5: The report feels too heavy with theoretical speculation on models for protein allostery and modularity. It's interesting discussion don't get me wrong, and belongs in the paper. However, it should be restricted to the discussion section so that it is more clear it is speculation. Plus, this approach would give more opportunity to speak to the result-driven evidence for how well the current sensors work and where their future utility lies.

Response: This is a good point and we have now focused the page 4 "Theoretical considerations" section on the experimental design. We moved the discussion on the biophysical and evolutionary forces controlling activity of the native and circularly permuted proteins in the discussion section.

Point 6: Did the authors try a non-permuted form of Nluc for their sensors? Curious if fusing the rapamycin (or similar) system to the native termini could work. If it failed or worked but not as well as the cp version this would give credence to the authors case for building the technology using a cp enzyme.

Response: This is a very good suggestion, and we regret not having performed this experiment for the original submission. We now constructed an analogue of rapamycin biosensor using wild type NanoLuc and demonstrate that it shows no appreciable change in its activity in response to rapamycin. These data are now presented in the figure S1D and described in the main paper pages 5 and 6.

On suggestion of the referee, we devised an experiment aimed to test whether the observed ligand-controlled activity switching was a consequence of circular permutation. To that end we fused FRB and FKBP to the N and C-termini of wild type NanoLuc via linkers of comparable length and tested the recombinantly produced protein for rapamycin-dependent activity (Table S1). Presence of rapamycin did not have an appreciable influence on the activity of fusion protein thus providing support to the role of circular permutation in the emergence of reversible local disorder (Fig. S1D).

Point 7 In "Construction of permuted NanoLuc luciferase" section of results it is implied that Renilla luciferase is ATP-dependent when it actually is not. Also in this section, NanoLuc luciferase is referred to as 'an engineered version of NanoLuc luciferase'. This doesn't make sense. It is assumed the authors mean to say that NanoLuc luciferase is an engineered enzyme derived from the native luciferase from Oplophorus. This should be updated to be more technically accurate.

Response: Corrected according to the suggestions of the referee.

Point 8 Figure 1 is not clear. It is hard to understand the various ribbon structures. Termini are difficult to see. Better color coating could be used to know where strands 9, 10 are in

panel C. The unstructured sequences could be labeled as strands 9, 10 to help with this. The authors could also consider adding a cartoon similar to what Dixon et al. used to describe the architecture of the split NanoLuc system (i.e., where strands 9, 10 are located).

Response: This is a good point and we now replaced Figures 1A and B with the annotated topology model.

Point 9. For the cp NanoLuc how is this sequence different from what Dixon et al published in 2016. They made a similar cp NanoLuc with a flexible linker connecting the former termini. If the sequence is as similar as it looks this could be made more clear and Dixon et al. should be cited.

Response: Please see the response to the point 2.

Point 10. For the cp NanoLuc what is the linker between former termini? Is it important to have a certain type of linker (e.g., hydrophobic, hydrophilic, flexible, rigid) or a precise linker length? Were these variables examined? If not could the system be further improved by targeting this region of the sequence?

Response: The exact sequence used for joining N- and C- termini can be found in the table S1. The construction of the biosensor is described on the top of the page 5 paragraph that we now modified to better explain the composition of the linker and to point the reader to the sequencing information.

To access two terminal β -strands we fused the native N and C-terminus of NanoLuc with a flexible linker composed of glycine serine repeats and reintroduced them in the loop connecting the last β -strand with the rest of the protein at position 161, resulting in formation of a circular permuted NanoLuc (cpNanoLuc) (Fig.1B, Table S1).

The structure and the length of the linker are expected to influence the performance of the biosensor such as the dynamic range, overall affinity for the ligand, the rate of response as well as maximal catalytic activity. However our initial experiments performed in response to the point 3 showed surprisingly little influence of N- and C-termini length on the core switch's background activity. Therefore, we are proposing that the sequence of the

terminal β -sheets is likely to play dominant role in the switches' behaviour. We have now added a following sentence to the discussion section reflecting this notion.

Our experiments did not find a strong correlation between the length and the structure of N- and C-terminal extensions and the background activity of the biosensor. While this is in line with our observation that structurally diverse ligand binding domains can be combined with the cpNanoLuc core module it also calls for further analysis of structural features determining its switchable behavior of cpNanoLuc. In particular, we expect that by modifying sequences of β -strands 1 and 10, as well as the length and composition of the linkers connecting the cpNanoLuc and binding domains it will be possible to modulate the dynamic range, the maximal catalytic activity and response times of the system.

Point 11. *For Figure 2 what is Calcineurin A and Calcineurin B? It feels like it is overly assumed that the reader will know what these are and how they relate to the rap system. Some elaboration would be helpful.*

Response: The description of tacrolimus biosensor codesign can be found on the page 7. We now expanded the description and refer the reader to the structure of Calcineurin A:Calcineurin B in complex with FKBP and FK506.

We have solved this problem by using the structure of quaternary complex of Calcineurin A and Calcineurin B in complex with FKBP and FK506 (PDB:1TCO) to design a linker between Calcineurin A and Calcineurin B converting it to a single polypeptide³¹. This polypeptide was used to replace the FRB domain in the rapamycin biosensor yielding a putative tacrolimus biosensor (Table S1).

Point 12. In figure 2 what are the raw RLU values? Could this be mentioned or added to supporting info? How does the luminescence intensity of the sensor compare to unmodified NanoLuc?

Response: The activity data of NanoLuc biosensors is presented in supporting information section as figure S3B. We now modified a sentence at the end of page 7 to point readers to this information.

Fully activated biosensors retained approximately 20% of wild type NanoLuc's luminescence and could be dried down and re-hydrated without optimization (Fig. S1E, 3B,C).

Point 13. *VHH domains refer to Nanobodies; is that correct? If so this could be made more clear. On a related note could the authors comment on how dependent their technology is on the existence of nanobodies or other binding moieties.*

Response: To make better connection between different terms used to describe camelid single chain antibody we now also mention and reference Nanobodies in the modified sentence on the page 8.

To this end we fused two VHH domains (also known as Nanobodies)³² recognizing two non-overlapping epitopes of α -amylase to both N and C-terminus of the developed biosensors.

Point 14. *Under the section "Functional mechanism of cpNanoLuc biosensors" what is the "activation delay" that the authors refer to? What is the magnitude of the delay and what is the perceived mechanism/explanation? How big a detriment is the delay?*

Response: The sentence is the reference to the data presented in the figure S3 that shows that dynamic range of the biosensor increases over time after exposure to the analyte reaching 50% in 5 minutes and 100% in 20 minutes. This is not an unusual feature of the artificial allosteric biosensors and may be improved upon. An example, for instance, is described in the provided reference 16. Understanding the structural rearrangements occurring in the biosensors as well as their kinetics was one of the motivations for performing HDX experiments. We, however, noticed that the biosensor activation data in supplementary materials section was not referenced in the main text and have now modified the sentence on the page 11 to point readers to that data.

Furthermore, it does not provide an explanation for the observed activation kinetics of the biosensor or guidance for its further improvement (Fig.S3).

Point 15: *The authors state their data indicates a greater stabilization of FKBP than FRB upon binding to rap. What can be said about the impact of rap binding on Nluc structure (independent of inducing proximity between strands 9, 10)?*

Response: We added the following sentence to the text to address the impact on stability in the NanoLuc domain of rapamycin binding to FKBP-FRB:

Stabilization of the NanoLuc domain is observed upon rap binding, but this effect is evidenced only at the longest HDX mixing time (3000 s), indicating the H-bond network of the two beta-sheets is intact, but involves dynamic instability in the apo form.

Point 16: Which strand are the authors referring to when they say that b-2 of the cpNanoLuc is detached from the b-sheet. Which strand is that? The original strand 1? And which b-sheet are they referring to, the original 10 that's now the N-term?

Response: This was an error that originated from the numbering of HDX data for the cpNanoLuc construct. We have altered this to read beta-10 in line with the nomenclature defined in Fig. 1.

Point 17: Figure 5 is very difficult to follow because there is such poor resolution with the structure images. Cartoons or even some form of text descriptions would be easier to follow. I suppose the figure could stand as is if moved to the supporting info.

Response: We believe that the data visualized on atomistic structural models is an important way for many readers to interpret our findings – in particular those in the structural biology community. However, we agree with the reviewer that the figure is a complex and detailed representation and the resolution may not be sufficient to show the features clearly in the main text. Therefore, we have replaced this version with one showing clear cartoon models that retain the key details and have moved the one containing atomistic protein surface structures to the Supplementary Information.

Reviewer #3

Point 1. Overstated claims: The authors have toned this down somewhat and do a somewhat better job of focusing on what is my view the most novel aspect, which is the ability to accommodate a wide range of 'binding domains' in this switch. Nonetheless, the advance reported here is still considerably less novel, in my opinion, than the authors are indicating, especially given their recent work on similar sensors. If the authors were genuinely setting out to prove their general hypothesis that circular permutants will often be

suitable for this type of sensor design, for example, then they would have had to generate several sensors from other circularly permuted proteins.

Response: We do not agree with the statement that this study is similar to our previously published work beyond the fact that they all aim to develop artificial allosteric proteins. Our previous studies used engineered auto-inhibition, protein splitting and domain insertion strategies to obtain switchable reported domains. To the best of our knowledge the concept of activity control through entropically driven local structural disorder presented in this study is distinct from other proposed mechanism and novel. To further strengthen the evidence of our model we have now added more experimental data (see response to referee 2 point 3 and referee 3 point 2) showing that circular permutation is critical for emergence of ligand controlled switchable behaviour. We feel that we now made a compelling case.

We agree with the referee that the demonstration of general applicability of such approach using different proteins is important. However, this goes far beyond the scope of this already very extensive study. In support of our claim, we include reference 47 to just published study where circular permutation of glucose dehydrogenase and its flanking with ligand binding domains leads to the emergence of switching properties. The conclusions of that study are in line with those presented here.

We believe that the next step would be to design a study where large libraries of circularly permuted mutants are created and analysed for ligand-dependence in order to understand the frequency with which such conditional allosteric system emerge. This notion is reflected in the statement made on the page 17.

Clearly, validation and assessment of the general nature of this phenomenon as well as the analysis of the governing forces (entropy, solvation and steric effects) cannot be performed using a single permutation and would require a quantitative analysis of a library of circularly permuted reporter domains and, possibly, a larger set of ligand binding domains. There is, however, accumulating evidence that the observed phenomenon is not unique to NanoLuc and other reporter domains could be converted into useful biosensors using a similar approach⁴⁷.

Point 2. HDX data: *The authors have also improved this section substantially, but now conduct a highly unusual (and I think misguided) analysis of the data. The authors appear*

*to be operating under the assumption that the midpoint of peptide-specific titration curves reflect the 'sensitivity' of certain regions to the presence of ligand. In fact, (assuming that k_{on} is large relative to the chemical exchange rate and $1/k_{off}$ is large relative to the labeling time, which is likely the case given the low K_d) there would be, at sub-stoichiometric concentrations of ligand, two uptakes for each (affected) peptide, one for the bound state and one for the unbound state, corresponding to distinct distributions with amplitudes linked to their relative populations. When these distributions have the same amplitude, (which should occur at just over 4 μM ligand for *every peptide*) then the ligand is 50% bound. Differences in the extent to which a given region is structurally impacted by ligand binding (i.e., the difference in uptake between the bound and unbound states) would impact the magnitude of the sigmoid, but not its mid-point, which would always be when the ligand is 50% bound... So why did the authors measure different EC_{50} s on different peptides? Possibly because of confounding factors like the rate at which conformational equilibrium is established in particular regions after the ligand binds (k_{on}) or unbinds (k_{off}) or non specific interactions with ligand (this second explanation is presented by the authors as a reason why most of their EC_{50} values don't make sense, given the low k_d of the ligand, which is a reasonable explanation, but it is another inexplicable thing about the data).*

I do think that the time-dependent changes in HDX uptake at stoichiometric+ concentrations of ligand more-or-less confirm that ligand binding induces restructuring of the protein, particularly in the N and C-terminal regions, and it would be far preferable, in my view, if the authors simply conducted this more conventional analysis.

Response: We agree that the modifications to this section have resulted in a much clearer presentation of the workflow and results. We make the same assumptions as the Reviewer in terms of k_{on} and k_{off} , as slow labeling times (≥ 30 s) were measured here and K_d for ligand was nM. The Reviewer goes on to accurately describe what would be expected to result from a two-state system (i.e. unbound/bound), where each peptide would share a global midpoint on the titration curve. It was indeed surprising that we observed a range of apparent midpoints for individual sites within the protein, but this is consistent with the presence of structured intermediates on the pathway between unbound-bound states. We interpret this similarly to multi-state protein (un)folding equilibria in which structured intermediates are differentially stabilised relative to each other under varying concentrations of a (de)stabilising agent. The concentration midpoint for the population of each

intermediate species will relate to their relative stability and the hydrogen-exchange of a particular peptide will be tightly linked to this, rather than to the evolution of the final bound state. We feel that this analysis offers important insight, but we agree that the limitations of the approach need to be more clearly highlighted. Therefore, we have updated the text on page 14 to include the following:

There are confounding factors that would complicate and potentially alter this interpretation, such as intermediates with higher or lower protection factors than the (un)bound states that are populated before or after on the pathway, or distortion of the apparent titration curve resulting from the HDX measurement error. Ultimately a kinetic model is preferred, though here that analysis would be under-determined. Nonetheless, allowing for the simplification that sequentially formed intermediate states of cpNanoLuc cannot reverse changes in HDX protection factor (for example a protected region does not subsequently become deprotected), this analysis provides considerable insight into the structural allosteric pathway of this biosensor.

Reviewers' Comments:

Reviewer #2:

Remarks to the Author:

1. I remain concerned the general tone around this technology is too speculative and bold as to its general utility and modularity. It's not that the technology and the work are not impressive, it's just that the implications still feel a bit overstated. It seems at least one other reviewer has similar feelings.

2. Figure 1 could still benefit from modification. First of all, panels a, b are cut and pasted from previous literature. It may be necessary to modify accordingly to avoid copyright issues?? (not sure).

The main concern is that panels c and d are still difficult to follow at a technical level. They're fine at a high level, i.e., rap binds to the sensor and the luciferase is refolded properly to give a functional enzyme. But the details are not clear. The model (minus Rap) indicates that the former beta-strand 10 (now on N-terminus of luciferase) is unstructured/unfolded and/or unable to complement. The former beta-strand 9 (now the new C-terminus) appears to be properly structured in 1a. As the new terminus aren't the authors arguing that it should be unstructured? In addition, it looks like according to the model in panel a that the former b-strand 1 (now beta-strand 2 in cp form) is unstructured. I'm not sure if this is what the authors are attempting to convey in their proposed mechanism for complementation.

3. As a follow-up to point 2, in terms of mechanism for the role of rap in complementation, wouldn't a change in proximity alone potentially be enough to foster complementation of functional enzyme

Reviewer #3:

Remarks to the Author:

This is the third review for this manuscript.

The authors have acknowledged the issues I identified with the HDX data and provide an explanation for their observations, albeit one that flies in the face of what is commonly understood about how ligand interactions impact conformational dynamics. Ligands are commonly understood to stabilize a particular conformation or conformational bias, not different conformations depending on the concentration of ligand. I suspect most people who work in protein folding/dynamics will find the author's explanation implausible if not non-sensical, and it is in any case only vaguely eluded to in the updated manuscript. My suggestion again would be to substantially reduce the 'fine' interpretation of the HDX data (which is not strictly necessary, in my view) and instead simply indicate that the data provide evidence for structural changes in the protein upon ligand binding.

The authors have attempted to address some of the issues of (potentially) overstated novelty raised by me and the other reviewers. It is admittedly challenging for the authors to fully address such issues, since they are innate to the reported work, and there is unquestionably some novelty and potential impact here. Ultimately, the issue of whether the novelty/expected impact is sufficient for the journal in question is properly addressed by the editor. From a technical point of view, I can only strongly suggest that the authors limit their interpretation of the HDX data to what it actually shows for sure, otherwise the work is technically sound in my view.

We addressed the specific criticism of the reviewers in the following way:

Reviewer #2

Point 1: I remain concerned the general tone around this technology is too speculative and bold as to its general utility and modularity. It's not that the technology and the work are not impressive, it's just that the implications still feel a bit overstated. It seems at least one other reviewer has similar feelings.

Response: We agree with the referee that our study provides no direct evidence to the general applicability of the presented approach. We modified the abstract, the introduction and the discussion sections to flag the need for testing the general nature of the phenomena through further comprehensive studies. We, however, point out that our results provide the motivation and a guidance in designing of such studies.

Abstract:

To our knowledge, this is the only fully integrated protein biosensor architecture suitable for detection of both small molecules and biological polymers. While a large study is required for determining how frequent synthetic allostery emerges following circular permutation and what types of protein folds are susceptible to it, this study provides motivation and justification for such efforts.

End of the introduction section:

We propose that this approach may be applied to other protein scaffolds, thereby representing a path to standardized approach to construction of artificial allosteric switches.

Page 18 of the discussion section:

Clearly, validation and assessment of the general nature of this phenomenon as well as the analysis of the governing forces (entropy, solvation and steric effects) cannot be performed using a single permutation and would require a quantitative analysis of a library of circular permuted reporter domains, a larger set of ligand binding domains as well as additional biophysical methods. There is, however, additional evidence that suggest that the observed phenomenon may not be unique to NanoLuc and other reporter domains could be converted into useful biosensors using a similar approach⁴⁸.

End of the discussion section:

In conclusion we propose that entropically driven local structural disorder may represent an alternative mechanism for construction of artificial allosteric systems with increased tolerance for the size and topology of the regulatory ligand. While much more comprehensive analysis is required to establish general applicability of this phenomenon, our results provide guidance on the design of such studies.

Point 2. Figure 1 could still benefit from modification. First of all, panels a, b are cut and pasted from previous literature. It may be necessary to modify accordingly to avoid copyright issues?? (not sure).

Response: The panels of the figure 1A and B were drawn by us from scratch and hence no copyright issues should arise. There is limited variation one can introduce into a protein topology drawing and hence they all look much the same. We now added labels to the alpha helices both to complete the annotation and to distinguish the figure from the previously published materials.

The main concern is that panels c and d are still difficult to follow at a technical level. They're fine at a high level, i.e., rap binds to the sensor and the luciferase is refolded properly to give a functional enzyme. But the details are not clear. The model (minus Rap) indicates that the former beta-strand 10 (now on N-terminus of luciferase) is unstructured/unfolded and/or unable to complement. The former beta-strand 9 (now the new C-terminus) appears to be properly structured in 1a. As the new terminus aren't the authors arguing that it should be unstructured? In addition, it looks like according to the model in panel a that the former beta-strand 1 (now beta-strand 2 in cp form) is unstructured. I'm not sure if this is what the authors are attempting to convey in their proposed mechanism for complementation.

Response: This valuable comment resulted in us revising figure 1 C and D. To make it easier for the reader to identify elements of secondary structure we now labelled them according to the figure 1D where we retained the nomenclature of the wild type of enzyme to avoid the confusion. We display the strand 10 as dislodged and disordered as this could be expected due to the insertion of a long unstructured linker connecting it to the rest of the molecule. The strand 9 that now forms the C-termini of the enzymatic domain is retained in its original position but displayed without secondary structure alluding to its less ordered state. We think that the figure 1 is now sufficiently well illustrates both the topology of wild type and cpNanoLuc and explains the proposed mechanistic model of emergent synthetic allostery.

Point 3. As a follow-up to point 2, in terms of mechanism for the role of rap in complementation, wouldn't a change in proximity alone potentially be enough to foster complementation of functional enzyme.

Response: According to the proposed model the proximity alone should be sufficient to reduce the hydrodynamic radius occupied by the disordered termini. This would favour native interactions of the termini with the rest of the protein and increase the occupancy of the catalytically active state. Again, we full heartedly agree with the earlier comments of the reviewer that a larger data set is needed to see how general this phenomenon is and what are the key parameters driving emergence and providing control of synthetic allostery.

Reviewer #3

The authors have acknowledged the issues I identified with the HDX data and provide an explanation for their observations, albeit one that flies in the face of what is commonly understood about how ligand interactions impact conformational dynamics. Ligands are commonly understood to stabilize a particular conformation or conformational bias, not different conformations depending on the concentration of ligand. I suspect most people who work in protein folding/dynamics will find the author's explanation implausible if not non-sensical, and it is in any case only vaguely eluded to in the updated manuscript. My suggestion again would be to substantially reduce the 'fine' interpretation of the HDX data (which is not strictly necessary, in my view) and instead simply indicate that the data provide evidence for structural changes in the protein upon ligand binding.

The authors have attempted to address some of the issues of (potentially) overstated novelty raised by me and the other reviewers. It is admittedly challenging for the authors to fully address such issues, since they are innate to the reported work, and there is unquestionably some novelty and potential impact here. Ultimately, the issue of whether the novelty/expected impact is sufficient for the journal in question is properly addressed by the editor. From a technical point of view, I can only strongly suggest that the authors limit their interpretation of the HDX data to what it actually shows for sure, otherwise the work is technically sound in my view.

Response:

We accept that the data presented here cannot categorically map the energy landscape and thus the postulated intermediates may be correlated structural changes within a single (active) state of the protein. It is, however, not our argument that different conformations are innately stabilized by different [ligand], but rather that a multi-state equilibrium will enrich (and then reduce) any on-pathway intermediate(s) as it traverses between populations of 100% initial and 100% final state, for example by increasing [ligand] or [denaturant]. We considered that the sub-saturating ligand concentrations may have enriched some conformations that we interpreted as either obligatory on-pathway intermediate states, or conformers within an active state.

We altered Figure 5 to remove the equilibrium pathway interpretation.

We altered the text on page 14 to limit our interpretation, to explicitly state the potential for artefacts to impact the analysis and to briefly include literature support for the widespread acceptance of multi-state pathways in both protein folding and receptor:ligand binding:

"To characterize the distinct structural transitions that occur at equilibrium from apo-protein to the mature active enzyme-rapamycin complex, we made a quantitative analysis of the HDX data in response to ligand and at each amino acid in the protein for which we had data (Fig. S7-8). We sought to cluster coherent conformational changes that are stabilized by ligand binding. To achieve this, we calculated the sum of observed deuterium labeling for the three mixing times at each rapamycin concentration. These were then fitted to a dose-response model (examples of unprocessed peptide level data shown in Fig. 4B and averaged per amino acid data shown in S8). This yielded two coefficients to the model fit (equation S2) – the EC50 midpoint and the Hill number, nH , which represent the sensitivity and cooperativity respectively of each amino acid to rapamycin ligand. By clustering the amino acids according to these attributes, we identified those parts of the protein that have correlated behavior in response to ligand (Fig. 4B). Examples from each of the seven clusters are shown in Table 1 (note cluster 6 indicates no detectable rapamycin-dependent response). It is possible that the observed range of EC50 values for amino acids that are perturbed upon rapamycin binding is an artefact of experimental factors, such as the limited D-labeling times. If real, then it would be indicative of a broad and rough energy landscape for conformers of the active ligand-bound state of the protein, as has been widely observed in natural systems, such as multi-state

functional selectivity in G-protein coupled receptors (GPCRs) and the folding of linear repeat proteins. There are confounding factors that would complicate and potentially alter this interpretation, such as conformers with higher or lower protection factors than the (un)bound states that are populated before or after on the pathway, or distortion of the apparent titration curve resulting from the HDX measurement error. Ultimately a kinetic model is preferred, though here that analysis would be under-determined. Nonetheless, this analysis provides considerable insight into the allosteric conformational changes of this biosensor. "